# Towards Unsupervised Training of Matching-based Graph Edit Distance Solver via Preference-aware GAN

**Wei Huang**
University of New South Wales
`w.c.huang@unsw.edu.au`

**Hanchen Wang**
University of Technology Sydney
`Hanchen.Wang@uts.edu.au`

**Dong Wen**
University of New South Wales
`dong.wen@unsw.edu.au`

**Shaozhen Ma**
University of New South Wales
`shaozhen.ma@unsw.edu.au`

**Wenjie Zhang**
University of New South Wales
`wenjie.zhang@unsw.edu.au`

**Xuemin Lin**
Shanghai Jiaotong University
`xuemin.lin@sjtu.edu.cn`

## Abstract

Graph Edit Distance (GED) is a fundamental graph similarity metric widely used in various applications. However, computing GED is an NP-hard problem. Recent state-of-the-art hybrid GED solver has shown promising performance by formulating GED as a bipartite graph matching problem, then leveraging a generative diffusion model to predict node matching between two graphs, from which both the GED and its corresponding edit path can be extracted using a traditional algorithm. However, such methods typically rely heavily on ground-truth supervision, where the ground-truth node matchings are often costly to obtain in real-world scenarios. In this paper, we propose GEDRanker, a novel unsupervised GAN-based framework for GED computation. Specifically, GEDRanker consists of a matching-based GED solver and introduces an interpretable preference-aware discriminator. By leveraging preference signals over different node matchings derived from edit path lengths, the discriminator can guide the matching-based solver toward generating high-quality node matching without the need for ground-truth supervision. Extensive experiments on benchmark datasets demonstrate that our GEDRanker enables the matching-based GED solver to achieve near-optimal solution quality without any ground-truth supervision. The source code is available at `https://github.com/piupiupiuu/GEDRanker`.

## 1 Introduction

Graph Edit Distance (GED) is a widely used graph similarity metric [1, 2, 3] that determines the minimum number of edit operations required to transform one graph into another. It has broad applications in domains such as pattern recognition [4, 5] and computer vision [6, 7]. Figure 1(a) illustrates an example of an optimal edit path for transforming $G_1$ to $G_2$ with $\text{GED}(G_1, G_2) = 4$. GED is particularly appealing due to its interpretability and its ability to capture both attribute-level and structural-level similarities between graphs. Traditional exact approaches for computing GED rely on A* search [8, 9, 10]. However, due to the NP-hardness of GED computation, these methods fail to scale to graphs with more than 16 nodes [9], making them impractical for large graphs.

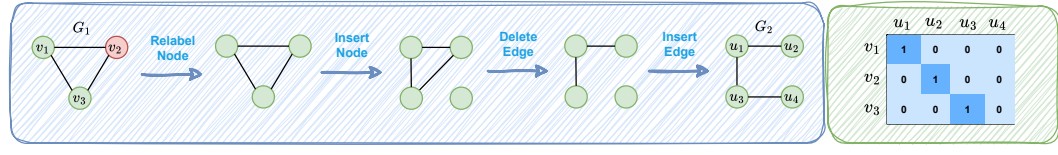

**(a) Optimal Edit Path**  **(b) Optimal Node Matching Matrix**

Figure 1: (a) An optimal edit path for converting $G_1$ to $G_2$ with $\text{GED}(G_1, G_2) = 4$. (b) An optimal node matching matrix from which an optimal edit path can be derived.

To address the limitations of traditional algorithms, hybrid approaches that combine deep learning with traditional algorithms have been widely explored. Recent hybrid approaches have focused on reformulating GED as a bipartite graph matching problem. Specifically, GEDGNN [11] and GEDIOT [12] propose to train a node matching model that predicts a single node matching probability matrix, where top-$k$ edit paths can be extracted sequentially from the predicted matrices using a traditional algorithm. Later on, the current state-of-the-art GED solver, DiffGED [13], employs a generative diffusion model to predict top-$k$ high quality node matching probability matrices in parallel, which parallelized the extraction of top-$k$ edit paths. Notably, DiffGED has achieved near-optimal solution quality, closely approximating the ground-truth GED with a remarkable computational efficiency. Unfortunately, all existing hybrid GED solvers heavily rely on supervised learning, requiring ground-truth node matching matrices and GED values, which are often infeasible to obtain in practice due to the NP-hard nature of GED computation. This limits the usability of existing GED methods in real-world scenarios. Therefore, developing an unsupervised training framework for GED solvers is crucial for enhancing real-world applicability. Despite its importance, this direction remains largely underexplored.

In this paper, we propose GEDRanker, a novel GAN-based framework that enables the unsupervised training of a matching-based GED solver without relying on ground-truth node matching matrices. Unlike existing unsupervised training techniques commonly used in other problems, such as REIN-FORCE [14], GEDRanker provides a more effective and interpretable learning objective (*i.e.,* loss function) that enables GED solver to directly optimize a GED-related score. Specifically, we design an efficient unsupervised training strategy that incorporates a preference-aware discriminator. Rather than directly estimating the resulting edit path length of the node matching matrix, our preference-aware discriminator evaluates the preference over different node matching matrices, thereby providing richer feedback to guide the matching-based GED solver toward exploring high-quality solutions.

**Contributions.** To the best of our knowledge, GEDRanker is the first framework that enables unsupervised training of supervised neural methods for GED computation, offering a scalable and label-free alternative to supervised approaches. Experimental results on benchmark datasets show that the current state-of-the-art supervised matching-based GED solver, when trained under our unsupervised framework, achieves near-optimal solution quality comparable to supervised training with the same number of epochs, and outperforms all other supervised learning methods and traditional GED solvers.

## 2  Related Work

**Traditional Approaches.**    Early approaches for solving GED are primarily based on exact combinatorial algorithms, such as A* search with handcrafted heuristics [9, 10]. While effective on small graphs, these methods are computationally prohibitive on larger graphs due to the NP-hardness of GED computation. To improve scalability, various approximate methods have been developed. These include bipartite matching formulations solved by the Hungarian algorithm [15] or the Volgenant-Jonker algorithm [16], as well as heuristic variants like A*-beam search [8] that limits the search space to enhance efficiency. However, such approximations often suffer from poor solution quality.

**Deep Learning Approaches.**    More recently, deep learning methods have emerged as promising alternatives. SimGNN [17] and its successors [18, 19, 20, 21, 22, 23] formulate GED estimation as a regression task, enabling fast prediction of GED values. However, these methods do not recover the edit path, which is often critical in real applications. As a result, the predicted GED can be smaller than the true value, leading to infeasible solutions for which no valid edit path exists.

**Hybrid Approaches.** To address the limitations of regression-based deep learning methods, hybrid frameworks have been extensively studied to jointly estimate GED and recover the corresponding edit path. A class of approaches, including Noah [24], GENN-A* [25], and MATA* [26], integrate GNNs with A* search by generating dynamic heuristics or candidate matches. However, these methods still inherit the scalability bottlenecks of A*-based search. To improve both solution quality and computational efficiency, GEDGNN [11] and GEDIOT [12] reformulate GED estimation as a bipartite node matching problem. In this setting, a node matching model is trained to predict a node matching probability matrix, from which top-$k$ edit paths are sequentially extracted using traditional algorithms. To further enhance solution quality and enable parallel extraction of top-$k$ candidates, DiffGED [13], employs a generative diffusion-based node matching model to predict top-$k$ node matching matrices in parallel, achieving near-optimal accuracy with significantly reduced running time. Motivated by its effectiveness and efficiency, we adopt the current state-of-the-art diffusion-based node matching model as the base GED solver of our unsupervised training framework.

## 3 Preliminaries

### 3.1 Problem Formulation

**Problem Definition (Graph Edit Distance Computation).** Given two graphs $G_1 = (V_1, E_1, L_1)$ and $G_2 = (V_2, E_2, L_2)$, where $V$, $E$ and $L$ represent the node set, edge set, and labeling function, respectively, we consider three types of edit operations: (1) Node insertion or deletion; (2) Edge insertion or deletion; (3) Node relabeling. An edit path is defined as a sequence of edit operations that transforms $G_1$ into $G_2$. The Graph Edit Distance (GED) is the length of the optimal edit path that requires the minimum number of edit operations. In this paper, we aim not only to predict GED, but also to recover the corresponding edit path for the predicted GED.

**Edit Path Generation.** Given a pair of graphs $G_1$ and $G_2$ (assuming $|V_1| \leq |V_2|$), let $\pi \in \{0, 1\}^{|V_1| \times |V_2|}$ denote a binary node matching matrix that satisfy the following constraint:

$$\sum_{u=1}^{|V_2|} \pi[v][u] = 1 \quad \forall v \in V_1, \quad \sum_{v=1}^{|V_1|} \pi[v][u] \leq 1 \quad \forall u \in V_2 \tag{1}$$

where $\pi[v][u] = 1$ if node $v \in V_1$ is matched to node $u \in V_2$; otherwise $\pi[v][u] = 0$. Each $v \in V_1$ matches to exactly one $u \in V_2$, and each $u \in V_2$ matches to at most one $v \in V_1$. Given $\pi$, an edit path can be derived with a linear time complexity of $O(|V_2| + |E_1| + |E_2|)$ as follows:

(1) For each node $u \in V_2$, if $u$ is matched to a node $v \in V_1$ and $L_1(v) \neq L_2(u)$, then relabel $L_1(v)$ to $L_2(u)$. If $u$ is not matched to any node, then insert a new node with label $L_2(u)$ into $V_1$, and match it to $u$. The overall time complexity of this step is $O(|V_2|)$.

(2) Suppose $v, v' \in V_1$ are matched to $u, u' \in V_2$, respectively. If $(v, v') \in E_1$ but $(u, u') \notin E_2$, then delete $(v, v')$ from $E_1$. If $(u, u') \in E_2$ but $(v, v') \notin E_1$, then insert $(v, v')$ into $E_1$. The overall time complexity of this step is $O(|E_1| + |E_2|)$.

Figure 1(b) shows an optimal node matching matrix from which the optimal edit path in Figure 1(a) is derived. Thus, computing GED can be reformulated as finding the optimal node matching matrix $\pi^*$ that minimizes the total number of edit operations $c(G_1, G_2, \pi^*)$ in the resulting edit path.

**Greedy Node Matching Matrix Decoding.** Let $\hat{\pi} \in [0, 1]^{|V_1| \times |V_2|}$ be a node matching probability matrix, where $\hat{\pi}[v][u]$ denotes the matching probability between node $v$ with node $u$. A binary node matching matrix $\pi$ can be greedily decoded from $\hat{\pi}$ with a time complexity of $O(|V_1|^2|V_2|)$ as follows:

(1) Select node pair $(v, u)$ with the highest probability in $\hat{\pi}$, and set $\pi[v][u] = 1$.

(2) Set all values in the $v$-th row and $u$-th column of $\hat{\pi}$ to $-\infty$.

(3) Repeat steps (1) and (2) for $|V_1|$ iterations.

Therefore, GED can be estimated by training a node matching model to generate node matching probability matrix that, when decoded, yields a node matching matrix minimizing the edit path length.

## 3.2 Supervised Diffusion-based Node Matching Model

The supervised diffusion-based node matching model [13] consists of a forward process and a reverse process. Intuitively, the forward process progressively corrupts the ground-truth node matching matrix $\pi^*$ over $T$ time steps to create a sequence of increasing noisy latent variables, such that $q(\pi^{1:T}|\pi^0) = \prod_{t=1}^{T} q(\pi^t|\pi^{t-1})$ with $\pi^0 = \pi^*$, where $\pi^t$ with $t > 0$ is the noisy node matching matrix that does not need to satisfy the constraint defined in Equation 1. Next, a denoising network $g_\phi$ takes as input a graph pair $(G_1, G_2)$, a noisy matching matrix $\pi^t$, and the corresponding time step $t$, is trained to reconstruct $\pi^*$ from $\pi^t$. During inference, the reverse process begins from a randomly sampled noisy matrix $\pi^{t_S}$, and iteratively applies $g_\phi$ to refine it towards a high-quality node matching matrix over a sequence of time steps $\{t_0, t_1, ..., t_S\}$ with $S \leq T$, $t_0 = 0$ and $t_S = T$, such that $p_\theta(\pi^{t_0:t_S}|G_1, G_2) = p(\pi^{t_S}) \prod_{i=1}^{S} p_\theta(\pi^{t_{i-1}}|\pi^{t_i}, G_1, G_2)$.

**Forward Process.** Specifically, let $\widetilde{\pi} \in \{0,1\}^{|V_1| \times |V_2| \times 2}$ denote the one-hot encoding of a node matching matrix $\pi \in \{0,1\}^{|V_1| \times |V_2|}$. The forward process corrupts $\pi^{t-1}$ to $\pi^t$ as: $q(\pi^t|\pi^{t-1}) = \mathrm{Cat}(\pi^t|p = \widetilde{\pi}^{t-1}Q_t)$, where $Q_t = \begin{bmatrix} 1 - \beta_t & \beta_t \\ \beta_t & 1 - \beta_t \end{bmatrix}$ is the transition probability matrix, Cat is the categorical distribution and $\beta_t$ denotes the corruption ratio. The $t$-step marginal can be written as: $q(\pi^t|\pi^0) = \mathrm{Cat}(\pi^t|p = \widetilde{\pi}^0 \overline{Q}_t)$, where $\overline{Q}_t = Q_1 Q_2 ... Q_t$, this allows $\pi^t$ to be sampled efficiently from $\pi^0$ during training.

**Reverse Process.** Starting from a randomly sampled noisy node matching matrix $\pi^{t_S}$, each step of the reverse process denoises $\pi^{t_i}$ to $\pi^{t_{i-1}}$ as follows:

$$q(\pi^{t_{i-1}}|\pi^{t_i}, \pi^0) = \frac{q(\pi^{t_i}|\pi^{t_{i-1}}, \pi^0)q(\pi^{t_{i-1}}|\pi^0)}{q(\pi^{t_i}|\pi^0)}$$
$$p_\phi(\widetilde{\pi}^0|\pi^{t_i}, G_1, G_2) = \sigma(g_\phi(G_1, G_2, \pi^{t_i}, t_i))$$
$$p_\phi(\pi^{t_{i-1}}|\pi^{t_i}, G_1, G_2) = \sum_{\widetilde{\pi}} q(\pi^{t_{i-1}}|\pi^{t_i}, \widetilde{\pi}^0)p_\phi(\widetilde{\pi}^0|\pi^{t_i}, G_1, G_2)$$
$$\pi^{t_{i-1}} \sim p_\phi(\pi^{t_{i-1}}|\pi^{t_i}, G_1, G_2)$$

(2)

where $q(\pi^{t_{i-1}}|\pi^{t_i}, \pi^0)$ is the posterior, $\sigma$ is the Sigmoid activation, and $p_\phi(\widetilde{\pi}^0|\pi^{t_i}, G, G')$ is the node matching probabilities predicted by $g_\phi$, the detailed architecture of $g_\phi$ is described in Appendix A.3.

During inference, for each reverse step except the final step, a binary noisy node matching matrix $\pi^{t_{i-1}}$ is sampled from $p_\phi(\pi^{t_{i-1}}|\pi^{t_i}, G_1, G_2)$ via Bernoulli sampling. For the final reverse step, we decode $p_\phi(\pi^0|\pi^{t_1}, G_1, G_2)$ following the greedy method described in Section 3.1 to obtain a constrained binary node matching matrix $\pi^0$. The detailed procedure of reverse process can be found in Appendix A.1. To enhance the solution quality, $k$ random initial $\pi^{t_S}$ could be sampled in parallel, and $k$ independent reverse processes could be performed in parallel to generate $k$ candidate node matching matrices, the one results in the shortest edit path will be the final solution.

**Supervised Training Strategy.** In the supervised learning setting, at each training step, given a pair of graphs, a random time step $t$ is sampled, and a noisy matching matrix $\pi^t$ is sampled from $q(\pi^t|\pi^0)$, where $\pi^0 = \pi^*$ is the ground-truth optimal node matching matrix. The denoising network $g_\phi$ then takes as input a graph pair $(G_1, G_2)$, a noisy matching matrix $\pi^t$, and the time step $t$, and is trained to recover $\pi^*$ from $\pi^t$ by minimizing:

$$\mathcal{L}_{rec(\pi^*)} = \frac{1}{|V_1||V_2|} \sum_{v=1}^{|V_1|} \sum_{u=1}^{|V_2|} (\pi^*[v][u] \log(\hat{\pi}_{g_\phi}[v][u]) + (1 - \pi^*[v][u]) \log(1 - \hat{\pi}_{g_\phi}[v][u])) \quad (3)$$

where $\hat{\pi}_{g_\phi}$ is the node matching probability matrix such that: $\hat{\pi}_{g_\phi} = \sigma(g_\phi(G_1, G_2, \pi^t, t))$. However, this training strategy heavily relies on pre-computation of the ground-truth node matching matrix.

## 4 Proposed Approach: GEDRanker

### 4.1 Unsupervised Training Strategy

In practice, the ground-truth $\pi^*$ is often unavailable, making it impractical to directly apply the supervised training strategy. Assuming the same set of training graphs as in the supervised learning

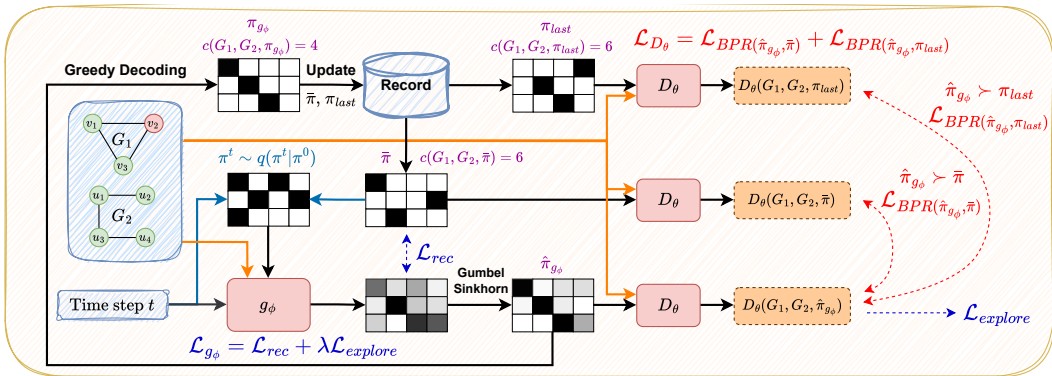

Figure 2: An overview of GEDRanker. For each training step, given a pair of training graphs, we maintain a record of the current best node matching matrix $\bar{\pi}$ and the node matching matrix obtained from the previous training step $\pi_{last}$. A noisy matching $\pi^t$ is sampled at a random diffusion time step $t$ and denoised by the denoising network $g_\phi$ to produce node matching scores. The resulting matching probability matrix $\hat{\pi}_{g_\phi}$ is obtained via Gumbel-Sinkhorn and greedily decoded to $\pi_{g_\phi}$. The preference-aware discriminator $D_\theta$ is trained to learn a preference ordering over $\bar{\pi}$, $\pi_{last}$, and $\hat{\pi}_{g_\phi}$. Next, $g_\phi$ is trained to recover $\bar{\pi}$ and maximize the preference score $D_\theta(G_1, G_2, \hat{\pi}_{g_\phi})$. Finally, the record is updated by $\pi_{g_\phi}$.

setting, a modified training strategy is to store a randomly initialized binary constrained node matching matrix as the current best solution $\bar{\pi}$ for each training graph pair. At each training step, for a given pair of graphs, we sample a random $t$, and sample a noisy matching matrix $\pi^t$ from $q(\pi^t|\pi^0)$, with $\pi^0 = \bar{\pi}$, then $g_\phi$ is trained to recover $\bar{\pi}$ from $\pi^t$ by minimizing $\mathcal{L}_{rec(\bar{\pi})}$. If the node matching matrix $\pi_{g_\phi}$ decoded from $\hat{\pi}_{g_\phi} = \sigma(g_\phi(G_1, G_2, \pi^t, t))$ yields a shorter edit path (i.e., $c(G_1, G_2, \pi_{g_\phi}) < c(G_1, G_2, \bar{\pi})$), then $\bar{\pi}$ is updated to $\pi_{g_\phi}$. This guarantees that $g_\phi$ progressively finds a better solution during training, while recovering the current best solution.

However, the training objective in the above approach is fundamentally exploitative, as it only focused on recovering the currently best solution rather than actively exploring alternative solutions. While exploitation is beneficial in later stages when a high-quality solution is available, it is inefficient in the early stages of training, where the current best solution is likely far from optimal. In such cases, blindly recovering a suboptimal node matching matrix is ineffective and can mislead $g_\phi$ into poor learning directions, and hinder its ability to explore better solutions. Therefore, an effective training objective should prioritize exploring for better solutions in the early stages and shift its focus toward recovering high-quality solutions in the later stages.

**RL-based Training Objective.** A widely adopted strategy for guiding $g_\phi$'s exploration towards better solutions is the use of reinforcement learning, specifically the REINFORCE [14], such that:

$$\nabla_\phi J(\phi) = \mathbb{E}_{\pi_{g_\phi}}[(c(G_1, G_2, \pi_{g_\phi}) - b)\nabla_\phi \log \sigma(g_\phi(\pi_{g_\phi}|G_1, G_2, \pi^t, t))] \tag{4}$$

where $c(G_1, G_2, \pi_{g_\phi})$ is the cost of the edit path computed according to the Edit Path Generation procedure in Section 3.1, and $b$ is a baseline used to reduce gradient variance. This method adjusts the parameters of $g_\phi$ to increase or decrease the matching probability of each node pair based on the resulting edit path length, encouraging the model to generate node matching probability matrices that are more likely to yield shorter edit paths.

Unfortunately, this approach suffers from several limitations: (1) It does not directly optimize the graph edit distance and lacks interpretability, as the edit path length only serves as a scale for gradient updates; (2) It assumes all node pairs in $\pi_{g_\phi}$ contribute equally to the edit path length, thus all node pairs are assigned with the same scale, making it difficult to distinguish correctly predicted node matching probabilities from incorrect ones; (3) It ignores the combinatorial dependencies among node pairs, but the generated edit path is highly sensitive to such dependencies, limiting $g_\phi$'s ability to efficiently explore high quality solutions.

**GAN-based Training Objective.** To overcome the limitations of REINFORCE, our GAN-based framework, GEDRanker, leverages a discriminator $D_\theta$ to guide $g_\phi$'s exploration for better solutions.

---

**Algorithm 1** Gumbel-Sinkhorn

---

**Input:** $g_\phi(G_1, G_2, \pi^t, t)$, number of iterations $K$, temperature $\tau$;
1: Sample Gumbel noise $Z \in \mathbb{R}^{|V_1| \times |V_2|}$ with $Z[v][u] \sim \text{Gumbel}(0,1)$;
2: $\hat{\pi}_{g_\phi} \leftarrow (g_\phi(G_1, G_2, \pi^t, t) + Z)/\tau$;
3: **for** $i = 1$ to $K$ **do**
4:     $\hat{\pi}_{g_\phi}[v][u] \leftarrow \hat{\pi}_{g_\phi}[v][u] - \log \sum_{u' \in V_2} \exp(\hat{\pi}_{g_\phi}[v][u'])$;        (row-wise normalization)
5:     $\hat{\pi}_{g_\phi}[v][u] \leftarrow \hat{\pi}_{g_\phi}[v][u] - \log \sum_{v' \in V_1} \exp(\hat{\pi}_{g_\phi}[v'][u])$;        (column-wise normalization)
6: **end for**
7: $\hat{\pi}_{g_\phi} \leftarrow \exp(\hat{\pi}_{g_\phi})$;
8: **return** $\hat{\pi}_{g_\phi}$;

---

Specifically, given a node matching matrix $\pi$, the discriminator $D_\theta$ is trained to evaluate $\pi$ and assign a GED-related score $D_\theta(G_1, G_2, \pi)$. If $\pi$ corresponds to a shorter edit path, then it will be assigned with a higher score; otherwise, a lower score. Therefore, $g_\phi$ is trained to maximize $D_\theta(G_1, G_2, \pi_{g_\phi})$.

However, the binary node matching matrix $\pi_{g_\phi}$ is greedily decoded from $\hat{\pi}_{g_\phi} = \sigma(g_\phi(G_1, G_2, \pi^t, t))$. This decoding process is non-differentiable, preventing $g_\phi$ from learning via gradient-based optimization. To address this issue, we employ the Gumbel-Sinkhorn method [27] to obtain a differentiable approximation of $\pi_{g_\phi}$ during training as outlined in Algorithm 1. Intuitively, Gumbel-Sinkhorn works by adding Gumbel noise to $g_\phi(G_1, G_2, \pi^t, t)$, followed by applying the Sinkhorn operator to produce a close approximation of a binary node matching matrix that satisfies the constraint defined in Equation 1. Let $S(\cdot)$ denote the Gumbel-Sinkhorn operation, we compute $\hat{\pi}_{g_\phi} = S(g_\phi(G_1, G_2, \pi^t, t))$ during training, and decode $\pi_{g_\phi}$ from $S(g_\phi(G_1, G_2, \pi^t, t))$ to generate the edit path, rather than decoding from $\sigma(g_\phi(G_1, G_2, \pi^t, t))$. Since $\hat{\pi}_{g_\phi}$ is now differentiable and closely approximates the decoded $\pi_{g_\phi}$, we can now train $g_\theta$ to maximize $D_\theta(G_1, G_2, \hat{\pi}_{g_\phi})$ by minimizing the following loss:

$$\mathcal{L}_{explore} = -D_\theta(G_1, G_2, \hat{\pi}_{g_\phi}) = -D_\theta(G_1, G_2, S(g_\phi(G_1, G_2, \pi^t, t))) \tag{5}$$

Through this GAN-based approach, $g_\phi$ can directly optimize a GED-related score and provides interpretability. Moreover, the discriminator $D_\theta$ can learn to capture the dependencies between node pairs, and distinguish the influence of each predicted value in $\hat{\pi}_{g_\phi}$ on the generated edit path, thus can guide the training of $g_\phi$ in a more efficient way. Overall, $g_\phi$ is trained to minimize:

$$\mathcal{L}_{g_\phi} = \mathcal{L}_{rec(\bar{\pi})} + \lambda \mathcal{L}_{explore} \tag{6}$$

where $\lambda$ is a hyperparameter that is dynamically decreased during training, encouraging exploration in the early stages and gradually shifting the focus toward exploitation in the later stages.

### 4.2 Discriminator

The discriminator $D_\theta$ is designed to assign a GED-related score to a given node matching matrix $\pi$ based on a graph pair. What score should $D_\theta$ output?

A natural approach to train $D_\theta$ is to directly predict the resulting edit path length of $\pi$ by minimizing the following loss:

$$\mathcal{L}_{\mathcal{D}_\theta} = (D_\theta(G_1, G_2, \pi) - \exp(-\frac{c(G_1, G_2, \pi) \times 2}{|V_1| + |V_2|}))^2 \tag{7}$$

where the edit path length is normalized by $\exp(-\frac{c(G_1, G_2, \pi) \times 2}{|V_1| + |V_2|}) \in (0, 1]$. Ideally, a shorter edit path should result in a higher score, allowing $g_\phi$ to learn to optimize the normalized GED. While this approach is straightforward and reasonable, can it guide $g_\phi$ towards the correct exploration direction?

Suppose we have two node matching matrices $\pi_1$ and $\pi_2$, with $\exp(-\frac{c(G_1, G_2, \pi_1) \times 2}{|V_1| + |V_2|}) = 0.4$ and $\exp(-\frac{c(G_1, G_2, \pi_2) \times 2}{|V_1| + |V_2|}) = 0.6$. Considering two cases, in Case 1, we have $D_\theta(G_1, G_2, \pi_1) = 0.1$ and $D_\theta(G_1, G_2, \pi_2) = 0.9$, resulting in $\mathcal{L}_{\mathcal{D}_\theta} = 0.18$. In Case 2, we have $D_\theta(G_1, G_2, \pi_1) = 0.6$ and $D_\theta(G_1, G_2, \pi_2) = 0.4$, resulting in $\mathcal{L}_{\mathcal{D}_\theta} = 0.08$. Clearly, Case 2 is preferred by $D_\theta$ due to the lower $\mathcal{L}_{\mathcal{D}_\theta}$. Consequently, based on the interpretable $\mathcal{L}_{explore}$, $g_\phi$ would be encouraged to generate $\pi_1$ over

$\pi_2$ as it results in a lower loss. However, $\pi_1$ actually corresponds to a longer edit path, meaning that the exploration direction of $g_\phi$ is misled by $D_\theta$, and such case would happen frequently when $D_\theta$ is not well trained. Thus, what matters for $D_\theta$ is not learning the precise normalized edit path length, but rather ensuring the correct preference ordering over different $\pi$.

**Preference-aware Training Objective.** To ensure that $D_\theta$ learns the correct preference, we introduce a preference-aware discriminator. A node matching matrix with a shorter edit path should be preferred ($\succ$) over one with a longer edit path. The objective of the preference-aware discriminator is to guarantee that the score assigned to the preferred node matching matrix is ranked higher than that of the less preferred one. Specifically, at each training step, given a pair of graphs, we have the current best node matching matrix $\bar{\pi}$ and the predicted node matching probability matrix $\hat{\pi}_{g_\phi} = S(g_\phi(G_1, G_2, \pi^t, t))$. The preference-aware discriminator is trained to minimize the following Bayes Personalized Ranking (BPR) loss [28]:

$$\mathcal{L}_{BPR(\hat{\pi}_{g_\phi}, \bar{\pi})} = \begin{cases} -\log \sigma(D_\theta(G_1, G_2, \hat{\pi}_{g_\phi}) - D_\theta(G_1, G_2, \bar{\pi})) & \text{if } c(G_1, G_2, \pi_{g_\phi}) \leq c(G_1, G_2, \bar{\pi}) \\ -\log \sigma(D_\theta(G_1, G_2, \bar{\pi}) - D_\theta(G_1, G_2, \hat{\pi}_{g_\phi})) & \text{if } c(G_1, G_2, \pi_{g_\phi}) \geq c(G_1, G_2, \bar{\pi}) \end{cases} \tag{8}$$

BPR encourages the preference-aware discriminator to increase the preference score of the better node matching matrix while simultaneously decreasing the preference score of the worse one, thereby promoting a clear separation between high and low quality solutions. Note that, other ranking loss functions, such as Hinge Loss, could be an alternative. Now, Case 1 yields a lower loss of $\mathcal{L}_{BPR(\pi_1, \pi_2)} = 0.3711$, whereas Case 2 results in a higher loss of $\mathcal{L}_{BPR(\pi_1, \pi_2)} = 0.7981$. Consequently, $D_\theta$ would choose Case 1, and $g_\phi$ would be encouraged to generate $\pi_2$ over $\pi_1$. Notably, for the special case where $c(G_1, G_2, \pi_{g_\phi}) = c(G_1, G_2, \bar{\pi})$, $\mathcal{L}_{BPR(\hat{\pi}_{g_\phi}, \bar{\pi})}$ is computed in both directions to ensure that the loss is minimized only if $\hat{\pi}_{g_\phi}$ and $\bar{\pi}$ are assigned with the same preference score.

Moreover, finding a better solution becomes increasingly challenging as training progresses. Consequently, $\bar{\pi}$ is updated less frequently in later training stages, leading to a scenario where a single node matching matrix may dominate $\bar{\pi}$. In this case, the discriminator can only learn to rank $\hat{\pi}_{g_\phi}$ against this specific $\bar{\pi}$, but may fail to correctly rank $\hat{\pi}_{g_\phi}$ against other node matching matrices. To deal with such case, we store an additional historical node matching matrix $\pi_{\text{last}}$ decoded from $\hat{\pi}_{g_\phi}$ in the previous training step. We then compute an additional ranking loss $\mathcal{L}_{BPR(\hat{\pi}_{g_\phi}, \pi_{\text{last}})}$ to encourage a more robust ranking mechanism. Overall, the discriminator is trained to minimize the following loss:

$$\mathcal{L}_{D_\theta} = \mathcal{L}_{BPR(\hat{\pi}_{g_\phi}, \bar{\pi})} + \mathcal{L}_{BPR(\hat{\pi}_{g_\phi}, \pi_{\text{last}})} \tag{9}$$

The overall framework of GEDRanker is illustrated in Figure 2 and outlined in Algorithm 3.

**Discriminator Architecture.** To effectively capture the dependencies among node pairs and distinguish the influence of each predicted node matching on the resulting edit path, our $D_\theta$ leverages GIN [29] and Anisotropic Graph Neural Network (AGNN) [30, 31, 32] to compute the embeddings of each node pair, then estimates the preference score directly based on these embeddings. Due to the space limitation, more details about the architecture of $D_\theta$ can be found in Appendix A.3 and C.3.

## 5 Experiments

### 5.1 Experimental Settings

**Datasets.** We conduct experiments on three widely used real world datasets: AIDS700 [17], Linux [33, 17], and IMDB [17, 34]. Each dataset is split into $60\%$, $20\%$ and $20\%$ as training graphs, validation graphs and testing graphs, respectively. We construct training, validation, and testing graph pairs, and generate their corresponding ground-truth GEDs and node matching matrices for evaluation following the strategy described in [11]. More details of the datasets can be found in Appendix B.1.

**Baselines.** We categorize baseline methods and our GEDRanker into three groups: (1) Traditional approximation methods: Hungarian [15], VJ [16] and GEDGW [12]; (2) Supervised hybrid methods: Noah [24], GENN-A* [25], MATA* [26], GEDGNN [11], GEDIOT [12] and DiffGED [13]; (3) Unsupervised hybrid method: GEDRanker. Due to the space limitations, the implementation details of our GEDRanker could be found in Appendix B.2.

Table 1: Overall performance on testing graph pairs. Methods with a running time exceeding 24 hours are marked with -. ↑: higher is better. ↓: lower is better. **Bold**: best in its own group. Trad, SL and UL denotes Traditional, Supervised Learning and Unsupervised Learning, respectively. Results for baselines, except for GEDIOT and GEDGW, are taken from [13].

| Datasets | Models | Type | MAE ↓ | Accuracy ↑ | $\rho$ ↑ | $\tau$ ↑ | p@10 ↑ | p@20 ↑ | Time(s) ↓ |
|---|---|---|---|---|---|---|---|---|---|
| AIDS700 | Hungarian | Trad | 8.247 | 1.1% | 0.547 | 0.431 | 52.8% | 59.9% | **0.00011** |
| | VJ | Trad | 14.085 | 0.6% | 0.372 | 0.284 | 41.9% | 52% | 0.00017 |
| | GEDGW | Trad | **0.811** | **53.9%** | **0.866** | **0.78** | **84.9%** | **85.7%** | 0.39255 |
| | Noah | SL | 3.057 | 6.6% | 0.751 | 0.629 | 74.1% | 76.9% | 0.6158 |
| | GENN-A* | SL | 0.632 | 61.5% | 0.903 | 0.815 | 85.6% | 88% | 2.98919 |
| | MATA* | SL | 0.838 | 58.7% | 0.8 | 0.718 | 73.6% | 77.6% | **0.00487** |
| | GEDGNN | SL | 1.098 | 52.5% | 0.845 | 0.752 | 89.1% | 88.3% | 0.39448 |
| | GEDIOT | SL | 1.188 | 53.5% | 0.825 | 0.73 | 88.6% | 86.7% | 0.39357 |
| | DiffGED | SL | **0.022** | **98%** | **0.996** | **0.992** | **99.8%** | **99.7%** | 0.0763 |
| | GEDRanker (Ours) | UL | **0.088** | **92.6%** | **0.984** | **0.969** | **99.1%** | **99.1%** | **0.0759** |
| Linux | Hungarian | Trad | 5.35 | 7.4% | 0.696 | 0.605 | 74.8% | 79.6% | **0.00009** |
| | VJ | Trad | 11.123 | 0.4% | 0.594 | 0.5 | 72.8% | 76% | 0.00013 |
| | GEDGW | Trad | **0.532** | **75.4%** | **0.919** | **0.864** | **90.5%** | **92.2%** | 0.1826 |
| | Noah | SL | 1.596 | 9% | 0.9 | 0.834 | 92.6% | 96% | 0.24457 |
| | GENN-A* | SL | 0.213 | 89.4% | 0.954 | 0.905 | 99.1% | 98.1% | 0.68176 |
| | MATA* | SL | 0.18 | 92.3% | 0.937 | 0.893 | 88.5% | 91.8% | **0.00464** |
| | GEDGNN | SL | 0.094 | 96.6% | 0.979 | 0.969 | 98.9% | 99.3% | 0.12863 |
| | GEDIOT | SL | 0.117 | 95.3% | 0.978 | 0.966 | 98.8% | 99% | 0.13535 |
| | DiffGED | SL | **0.0** | **100%** | **1.0** | **1.0** | **100%** | **100%** | 0.06982 |
| | GEDRanker (Ours) | UL | **0.01** | **99.5%** | **0.997** | **0.995** | **100%** | **99.8%** | **0.06973** |
| IMDB | Hungarian | Trad | 21.673 | 45.1% | 0.778 | 0.716 | 83.8% | 81.9% | **0.0001** |
| | VJ | Trad | 44.078 | 26.5% | 0.4 | 0.359 | 60.1% | 62% | 0.00038 |
| | GEDGW | Trad | **0.349** | **93.9%** | **0.966** | **0.953** | **99.1%** | **98.3%** | 0.37496 |
| | Noah | SL | - | - | - | - | - | - | - |
| | GENN-A* | SL | - | - | - | - | - | - | - |
| | MATA* | SL | - | - | - | - | - | - | - |
| | GEDGNN | SL | 2.469 | 85.5% | 0.898 | 0.879 | 92.4% | 92.1% | 0.42428 |
| | GEDIOT | SL | 2.822 | 84.5% | 0.9 | 0.878 | 92.3% | 92.7% | 0.41959 |
| | DiffGED | SL | **0.937** | **94.6%** | **0.982** | **0.973** | **97.5%** | **98.3%** | **0.15105** |
| | GEDRanker (Ours) | UL | **1.019** | **94%** | **0.999** | **0.97** | **96.1%** | **97%** | **0.15111** |

**Evaluation Metrics.** We evaluate the performance of each model using the following metrics: (1) *Mean Absolute Error* (MAE) measures average absolute error between the predicted GED and the ground-truth GED; (2) *Accuracy* measures the proportion of test pairs whose predicted GED equals the ground-truth GED; (3) *Spearman's Rank Correlation Coefficient* ($\rho$), and (4) *Kendall's Rank Correlation Coefficient* ($\tau$) both measure the matching ratio between the ranking results of the predicted GED and the ground-truth GED for each query test graph; (5) *Precision at* 10/20 ($p@10$, $p@20$) measures the proportion of predicted top-10/20 most similar graphs that appear in the ground-truth top-10/20 similar graphs for each query test graph. (6) *Time (s)* measures the average running time of each test graph pair.

## 5.2 Main Results

Table 1 presents the overall performance of each method on the test graph pairs. Among supervised methods, the diffusion-based node matching model DiffGED achieves near-optimal accuracy across all datasets. Our unsupervised GEDRanker, which trains diffusion-based node matching model without ground-truth labels but with the same number of training epochs, also outperforms other supervised methods and achieves near-optimal solution quality, with an insignificant performance gap compared to DiffGED.

Furthermore, compared to the traditional approximation method GEDGW, GEDRanker consistently achieves higher solution quality while maintaining shorter running times across all datasets. It is worth noting that on the IMDB dataset, GEDGW exhibits a lower MAE, this is because IMDB contains synthetic large graph pairs, where approximated ground-truth GED values might be higher

Table 2: Ablation study.

| Datasets | Models | MAE $\downarrow$ | Accuracy $\uparrow$ | $\rho \uparrow$ | $\tau \uparrow$ | $p@10 \uparrow$ | $p@20 \uparrow$ |
|---|---|---|---|---|---|---|---|
| AIDS700 | GEDRanker (plain) | 0.549 | 65.4% | 0.905 | 0.837 | 91.7% | 91.5% |
| | GEDRanker (RL) | 0.458 | 69.3% | 0.921 | 0.86 | 91.1% | 92.7% |
| | GEDRanker (GED) | 0.237 | 82.3% | 0.956 | 0.919 | 97.6% | 96.5% |
| | GEDRanker (Hinge) | 0.114 | 90.5% | 0.98 | 0.96 | 98.9% | 98.6% |
| | **GEDRanker** | **0.088** | **92.6%** | **0.984** | **0.969** | **99.1%** | **99.1%** |
| Linux | GEDRanker (plain) | 0.079 | 96.2% | 0.984 | 0.973 | 98.1% | 98.3% |
| | GEDRanker (RL) | 0.04 | 98% | 0.99 | 0.984 | 99.2% | 99.2% |
| | GEDRanker (GED) | 0.017 | 99.2% | 0.995 | 0.992 | 99.4% | 99.6% |
| | GEDRanker (Hinge) | 0.002 | 99.9% | 0.999 | 0.999 | 100% | 100% |
| | **GEDRanker** | **0.01** | **99.5%** | **0.997** | **0.995** | **100%** | **99.8%** |

than the actual ground-truth GED values. As a result, a lower MAE only reflects the predicted GED values are close to the approximated ground-truth values.

### 5.3 Ablation Study

**Exploration Ability.** To evaluate the significant of the exploration during unsupervised learning as well as the exploration ability of our proposed GEDRanker, we create three variant models: (1) *GEDRanker (plain)*, which simply trains $g_\phi$ only to recover the current best solution using $\mathcal{L}_{rec(\bar{\pi})}$ without any exploration; (2) *GEDRanker (RL)*, which guides $g_\phi$'s exploration using REINFORCE by replacing $\mathcal{L}_{explore}$ with Equation 4; (3) *GEDRanker (GED)*, which replaces the preference-aware discriminator with a discriminator that directly estimates the edit path length (Equation 7).

Table 2 presents the performance of each variant model on testing graph pairs. The results show that the performance of GEDRanker (plain) drops significantly without exploration, indicating that a purely exploitative approach focused solely on recovering the current best solution is insufficient. Furthermore, Figure 3 shows the average edit path length of the best found solutions on training graph pairs. Obviously, GEDRanker (plain) exhibits the weakest ability to discover better solutions, which misleads $g_\phi$ into recovering suboptimal solutions.

For the variant models that incorporate exploration, GEDRanker (RL), which guides the exploration of $g_\phi$ using REINFORCE, slightly improves the overall performance and exploration ability compared to GEDRanker (plain) as demonstrated in Table 2 and Figure 3. However, there still remains a significant performance gap compared to our GAN-based approaches GEDRanker and GEDRanker (GED). This gap arises because REINFORCE cannot directly optimize GED and struggles to capture the dependencies among node pairs as well as the individual impact of each node pair on the resulting edit path, whereas our GAN-based framework is able to capture both effectively.

Moreover, the preference-aware discriminator used in GEDRanker further enhances the exploration ability, compared to the discriminator used in GEDRanker (GED) that directly estimates the edit path length. As shown in Figure 3, the average edit path length of the best solution found by GEDRanker converges to a near-optimal value significantly faster than all variant models, indicating a strong ability to explore for high quality solutions. Consequently, this allows $g_\phi$ more epochs to recover high quality solutions, thereby improving the overall performance, as demonstrated in Table 2.

In contrast, GEDRanker (GED) struggles to identify better solutions during the early training phase. This observation aligns with our earlier analysis: a discriminator that directly estimates edit path length may produce incorrect preferences over different $\pi$ when $D_\theta$ is not yet well-trained, thereby misguiding the exploration direction of $g_\phi$. As training progresses, the edit path

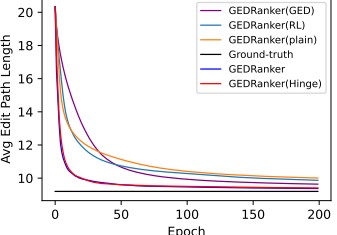

(a) AIDS700

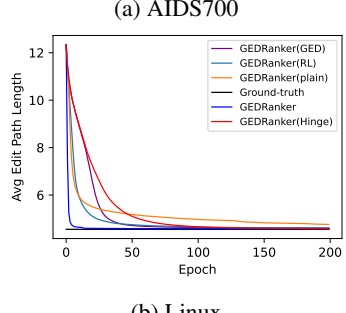

(b) Linux

Figure 3: Average edit path length of the best found node matching matrices on training graph pairs.

estimation becomes more accurate, such incorrect preference occurs less frequently, thus its exploration ability becomes better than GEDRanker (plain) and GEDRanker (RL).

**Ranking Loss.** For the preference-aware discriminator, we adopt the BPR ranking loss. To evaluate the effect of alternative ranking loss, we introduce a variant model, *GEDRanker (Hinge)*, which replaces BPR with a Hinge loss to rank the preference over different node matching matrices $\pi$. Following the same setting as in Equation 8, the Hinge loss is computed with a margin of 1 when the two node matching matrices yield different edit path lengths, and with a margin of 0 when they result in the same edit path length. In the latter case, the loss is computed in both directions to encourage their predicted scores to be as similar as possible. The detailed Hinge loss is shown in Appendix A.4.

The overall performance of GEDRanker (Hinge) is very close to that of GEDRanker with BPR loss, and both outperform all non-preference-aware variants, as shown in Table 2, further validating the effectiveness of our proposed preference-aware discriminator. However, as shown in Figure 3, the exploration ability of GEDRanker (Hinge) is less stable compared to GEDRanker. On the AIDS700 dataset, they both exhibit similar exploration trends, while on the Linux dataset, GEDRanker (Hinge) performs poorly during the early training phase. This discrepancy arises because Hinge loss produces non-smooth, discontinuous gradients, and it is sensitive to the choice of margin, making optimization less stable. Therefore, we choose to adopt BPR loss for our preference-aware discriminator.

# 6   Conclusion

In this paper, we propose GEDRanker, the first unsupervised training framework for GED computation. Specifically, GEDRanker adopts a GAN-based design, consisting of a node matching-based GED solver and a preference-aware discriminator that evaluates the relative quality of different node matching matrices. Extensive experiments on benchmark datasets demonstrate that GEDRanker enables the current state-of-the-art supervised GED solver to achieve near-optimal performance under a fully unsupervised setting, while also outperforming all other supervised and traditional baselines.

## Acknowledgments

Hanchen Wang is supported by ARC DE250100226. Dong Wen is supported by ARC DP230101445 and ARC DE240100668.

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

Figure 4: Reverse process of diffusion-based node matching model during inference.

# Appendix

# A  Detailed Method

---
**Algorithm 2** Reverse Process
---

**Input:** A pair of graphs $(G_1, G_2)$;
1: $\pi^{t_S} \sim \text{Bernoulli}(0.5)^{|V_1| \times |V_2|}$;
2: **for** $i = S$ to 1 **do**
3:     $p_\phi(\widetilde{\pi}^0 | \pi^{t_i}, G_1, G_2) \leftarrow \sigma(g_\phi(G_1, G_2, \pi^{t_i}, t_i))$;
4:     **if** $i \neq 1$ **then**
5:         $\pi^{t_{i-1}} \sim p_\phi(\pi^{t_{i-1}} | \pi^{t_i}, G_1, G_2)$;
6:     **else**
7:         $\pi^0 \leftarrow Greedy(p_\phi(\pi^{t_{i-1}} | \pi^{t_i}, G_1, G_2))$;
8:     **end if**
9: **end for**
10: **return** $\pi^0$;

---

---
**Algorithm 3** GEDRanker Training Strategy
---

1: Initialize $\bar{\pi} \leftarrow Greedy(\hat{\pi})$, $\pi_{\text{last}} \leftarrow Greedy(\hat{\pi})$, with $\hat{\pi} \sim \mathcal{U}(0,1)^{|V_1| \times |V_2|}$;
2: **for** each epoch **do**
3:     Sample $t \sim \mathcal{U}(\{1, ..., T\})$ and $\pi^t \sim q(\pi^t | \bar{\pi})$;
4:     $\hat{\pi}_{g_\phi} \leftarrow S(g_\phi(G_1, G_2, \pi^t, t))$;
5:     $\pi_{g_\phi} \leftarrow Greedy(\hat{\pi}_{g_\phi})$;
6:     Compute $D_\theta(G_1, G_2, \hat{\pi}_{g_\phi})$, $D_\theta(G_1, G_2, \bar{\pi})$, $D_\theta(G_1, G_2, \pi_{\text{last}})$;
7:     Compute $\mathcal{L}_{\mathcal{D}_\theta}$ and update $D_\theta$;
8:     Compute $D_\theta(G_1, G_2, \hat{\pi}_{g_\phi})$;
9:     Compute $\mathcal{L}_{g_\phi}$ and update $g_\phi$;
10:     **if** $c(G_1, G_2, \pi_{g_\phi}) < c(G_1, G_2, \bar{\pi})$ **then**
11:         Update $\bar{\pi} \leftarrow \pi_{g_\phi}$;
12:     **end if**
13:     Update $\pi_{\text{last}} \leftarrow \pi_{g_\phi}$;
14: **end for**

---

## A.1  Diffusion-based Node Matching Model Reverse Process

The detailed reverse process of diffusion-based node matching model is illustrated in Figure 4 and outlined in Algorithm 2.

## A.2  GEDRanker

The detailed unsupervised training strategy of GEDRanker is outlined in Algorithm 3.

## A.3  Network Architecture

**Network Architecture of $g_\phi$.**   The denoising network $g_\phi$ takes as input a graph pair $(G_1, G_2)$, a noisy matching matrix $\pi^t$, and the corresponding time step $t$, then predicts a matching score for each node pair. Intuitively, $g_\phi$ works by computing pair embedding of each node pair, followed by computing the matching score of each node pair based on its pair embedding.

Specifically, for each node $v \in V_1$ and $u \in V_2$, we initialize the node embeddings $\boldsymbol{h}_v^0$ and $\boldsymbol{h}_u^0$ as the one-hot encoding of their labels $L_1(v)$ and $L_2(u)$, respectively. For each node pair $(v, u) \in \pi^t$, we construct two directional pair embeddings $\boldsymbol{h}_{vu}^0$ and $h_{uv}^0$ to ensure permutation invariance, since the input graph pair has no inherent order, i.e., $\text{GED}(G_1, G_2) = \text{GED}(G_2, G_1)$. Both $\boldsymbol{h}_{vu}^0$ and $\boldsymbol{h}_{uv}^0$ are initialized using the sinusodial embeddings [35] of $\pi^t[v][u]$. Moreover, the time step embedding $h_t$ is initialized as the sinusodial embedding of $t$.

Each layer $l$ of $g_\phi$ consists of a two stage update. In the first stage, we independently update the node embeddings of each graph based on its own graph structure using siamese GIN [29] as follows:

$$
\begin{aligned}
\hat{\boldsymbol{h}}_v^l &= \text{GN}_{G_1}(\text{MLP}((1 + \epsilon^l) \cdot \boldsymbol{h}_v^{l-1} + \sum_{v' \in \mathcal{N}_{G_1}(v)} \boldsymbol{h}_{v'}^{l-1})) \\
\hat{\boldsymbol{h}}_u^l &= \text{GN}_{G_2}(\text{MLP}((1 + \epsilon^l) \cdot \boldsymbol{h}_u^{l-1} + \sum_{u' \in \mathcal{N}_{G_2}(u)} \boldsymbol{h}_{u'}^{l-1}))
\end{aligned}
\tag{10}
$$

where $\epsilon^l$ is a learnable scalar, MLP denotes a multi-layer perceptron, $\text{GN}_{G_1}$ and $\text{GN}_{G_2}$ denote graph normalization [36] over $G_1$ and $G_2$, respectively.

In the second stage, we simultaneously update the node embeddings of both graphs and the pair embeddings via Anisotropic Graph Neural Network (AGNN) [30, 31, 32], which incorporates the noisy interactions between node pairs and the corresponding time step $t$:

$$
\begin{aligned}
\hat{\boldsymbol{h}}_{vu}^l &= \boldsymbol{W}_1^l \boldsymbol{h}_{vu}^{l-1}, \quad \hat{\boldsymbol{h}}_{uv}^l = \boldsymbol{W}_1^l \boldsymbol{h}_{uv}^{l-1} \\
\tilde{\boldsymbol{h}}_{vu}^l &= \boldsymbol{W}_2^l \hat{\boldsymbol{h}}_{vu}^l + \boldsymbol{W}_3^l \hat{\boldsymbol{h}}_v^l + \boldsymbol{W}_4^l \hat{\boldsymbol{h}}_u^l \\
\tilde{\boldsymbol{h}}_{uv}^l &= \boldsymbol{W}_2^l \hat{\boldsymbol{h}}_{uv}^l + \boldsymbol{W}_3^l \hat{\boldsymbol{h}}_u^l + \boldsymbol{W}_4^l \hat{\boldsymbol{h}}_v^l \\
\boldsymbol{h}_{vu}^l &= \hat{\boldsymbol{h}}_{vu}^l + \text{MLP}(\text{ReLU}(\text{GN}_\pi(\tilde{\boldsymbol{h}}_{vu}^l)) + \boldsymbol{W}_5^l h_t) \\
\boldsymbol{h}_{uv}^l &= \hat{\boldsymbol{h}}_{uv}^l + \text{MLP}(\text{ReLU}(\text{GN}_\pi(\tilde{\boldsymbol{h}}_{uv}^l)) + \boldsymbol{W}_5^l h_t) \\
\boldsymbol{h}_v^l &= \hat{\boldsymbol{h}}_v^l + \text{ReLU}(\text{GN}_{G_1 G_2}(\boldsymbol{W}_6^l \hat{\boldsymbol{h}}_v^l + \sum_{u' \in V_2} \boldsymbol{W}_7^l \hat{\boldsymbol{h}}_{u'}^l \odot \sigma(\tilde{\boldsymbol{h}}_{vu'}^l))) \\
\boldsymbol{h}_u^l &= \hat{\boldsymbol{h}}_u^l + \text{ReLU}(\text{GN}_{G_1 G_2}(\boldsymbol{W}_6^l \hat{\boldsymbol{h}}_u^l + \sum_{v' \in V_1} \boldsymbol{W}_7^l \hat{\boldsymbol{h}}_{v'}^l \odot \sigma(\tilde{\boldsymbol{h}}_{uv'}^l)))
\end{aligned}
\tag{11}
$$

where $\boldsymbol{W}_1^l, \boldsymbol{W}_2^l, \boldsymbol{W}_3^l, \boldsymbol{W}_4^l, \boldsymbol{W}_5^l, \boldsymbol{W}_6^l, \boldsymbol{W}_7^l$ are learnable parameters at layer $l$, $\text{GN}_\pi$ denote the graph normalization over all node pairs and $\text{GN}_{G_1 G_2}$ denote the graph normalization over all nodes in both graphs $G_1$ and $G_2$.

For a $L$-layer denoising network $g_\phi$, the final matching score of each node pair is computed as follows:

$$
g_\phi(G_1, G_2, \pi^t, t)[v][u] = \text{MLP}(\boldsymbol{h}_{vu}^L) + \text{MLP}(\boldsymbol{h}_{uv}^L)
\tag{12}
$$

**Network Architecture of $D_\theta$.**   The discriminator $D_\theta$ takes as input a graph pair $(G_1, G_2)$ and a matching matrix $\pi$, then predicts a overall score for $\pi$. To better evaluate $\pi$, $D_\theta$ adopts the same structure as $g_\phi$, which computes the embeddings of each node pair in $\pi$, and computes the overall score based on the pair embeddings. Specifically, computing the embeddings for each node pair enables $D_\theta$ to distinguish the influence of individual node pairs in $\pi$, while the use of AGNN allows $D_\theta$ to effectively capture complex dependencies between node pairs. An overview of $D_\theta$'s network architecture is presented in Figure 5(a).

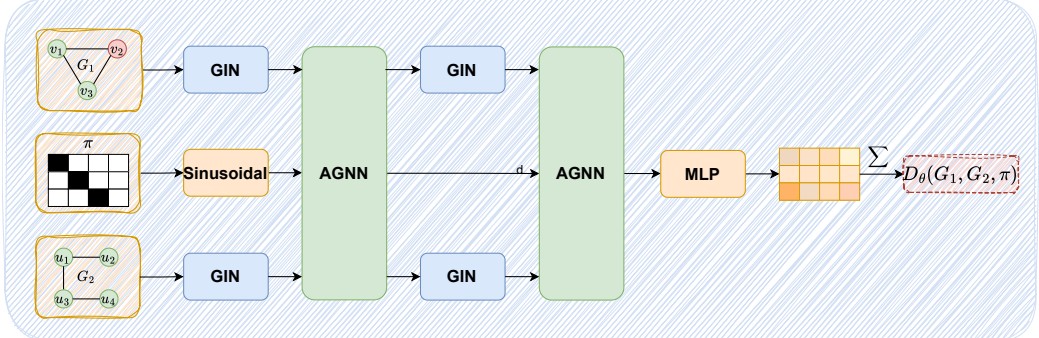

(a) Network architecture of $D_\theta$ in GEDRanker

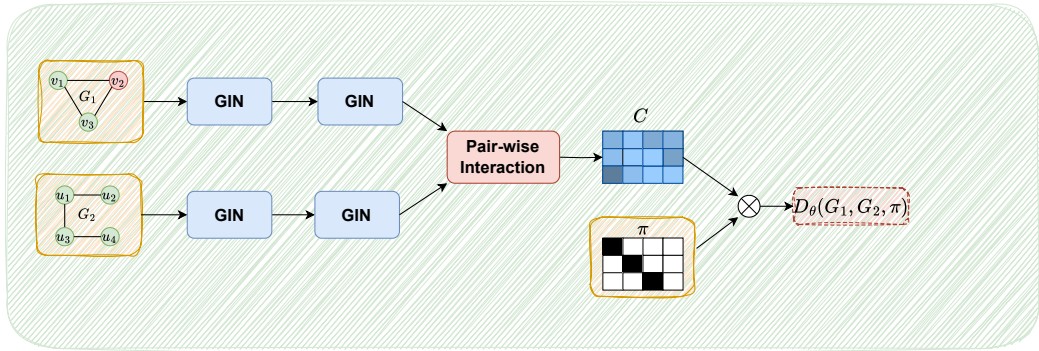

(b) Network architecture of $D_\theta$ in GEDRanker (cost)

Figure 5: Comparison of $D_\theta$'s network architecture.

However, since $D_\theta$ does not include the time step $t$ as an input, we remove the time step component from AGNN of the second update stage as follows:

$$
\begin{aligned}
\hat{\boldsymbol{h}}_{vu}^l &= \boldsymbol{W}_1^l \boldsymbol{h}_{vu}^{l-1}, \quad \hat{\boldsymbol{h}}_{uv}^l = \boldsymbol{W}_1^l \boldsymbol{h}_{uv}^{l-1} \\
\tilde{\boldsymbol{h}}_{vu}^l &= \boldsymbol{W}_2^l \hat{\boldsymbol{h}}_{vu}^l + \boldsymbol{W}_3^l \hat{\boldsymbol{h}}_v^l + \boldsymbol{W}_4^l \hat{\boldsymbol{h}}_u^l \\
\tilde{\boldsymbol{h}}_{uv}^l &= \boldsymbol{W}_2^l \hat{\boldsymbol{h}}_{uv}^l + \boldsymbol{W}_3^l \hat{\boldsymbol{h}}_u^l + \boldsymbol{W}_4^l \hat{\boldsymbol{h}}_v^l \\
\boldsymbol{h}_{vu}^l &= \hat{\boldsymbol{h}}_{vu}^l + \mathrm{MLP}(\mathrm{ReLU}(\mathrm{GN}_\pi(\tilde{\boldsymbol{h}}_{vu}^l))) \\
\boldsymbol{h}_{uv}^l &= \hat{\boldsymbol{h}}_{uv}^l + \mathrm{MLP}(\mathrm{ReLU}(\mathrm{GN}_\pi(\tilde{\boldsymbol{h}}_{uv}^l))) \\
\boldsymbol{h}_v^l &= \hat{\boldsymbol{h}}_v^l + \mathrm{ReLU}(\mathrm{GN}_{G_1 G_2}(\boldsymbol{W}_6^l \hat{\boldsymbol{h}}_v^l + \sum_{u' \in V_2} \boldsymbol{W}_7^l \hat{\boldsymbol{h}}_{u'}^l \odot \sigma(\tilde{\boldsymbol{h}}_{vu'}^l))) \\
\boldsymbol{h}_u^l &= \hat{\boldsymbol{h}}_u^l + \mathrm{ReLU}(\mathrm{GN}_{G_1 G_2}(\boldsymbol{W}_6^l \hat{\boldsymbol{h}}_u^l + \sum_{v' \in V_1} \boldsymbol{W}_7^l \hat{\boldsymbol{h}}_{v'}^l \odot \sigma(\tilde{\boldsymbol{h}}_{uv'}^l)))
\end{aligned}
\tag{13}
$$

For a $L$-layer discriminator $D_\theta$, the overall score of $\pi$ is computed as follows:

$$
D_\theta(G_1, G_2, \pi) = \sum_{(v,u) \in \pi} (\mathrm{MLP}(\boldsymbol{h}_{vu}^L) + \mathrm{MLP}(\boldsymbol{h}_{uv}^L))
\tag{14}
$$

## A.4 Alternative Ranking Loss

**Hinge Loss.** Hinge loss could be an alternative ranking loss for our preference-aware discriminator. At each training step, for a pair of graphs, given the current best node matching matrix $\bar{\pi}$ and the

predicted $\hat{\pi}_{g_\phi}$, the discriminator can be trained to minimize the following Hinge loss:

$$\mathcal{L}_{Hinge(\hat{\pi}_{g_\phi}, \bar{\pi})} = \begin{cases} \max(0, D_\theta(G_1, G_2, \bar{\pi}) - D_\theta(G_1, G_2, \hat{\pi}_{g_\phi}) + \text{margin}) & \text{if } c(G_1, G_2, \pi_{g_\phi}) \leq c(G_1, G_2, \bar{\pi}) \\ \max(0, D_\theta(G_1, G_2, \hat{\pi}_{g_\phi}) - D_\theta(G_1, G_2, \bar{\pi}) + \text{margin}) & \text{if } c(G_1, G_2, \pi_{g_\phi}) \geq c(G_1, G_2, \bar{\pi}) \end{cases} \tag{15}$$

where margin is set to 0 if $c(G_1, G_2, \pi_{g_\phi}) = c(G_1, G_2, \bar{\pi})$, otherwise 1.

# B Experimental Setting

## B.1 Dataset

We conduct experiments on three widely used real world datasets: AIDS700 [17], Linux [33, 17], and IMDB [17, 34]. Table 3 shows the statistics of the datasets. For each dataset, we split into 60%, 20% and 20% as training graphs, validation graphs and testing graphs, respectively.

To construct training graph pairs, all training graphs with no more than 10 nodes are paired with each other. For these small graph pairs, the ground-truth GEDs and node matching matrices are obtained using the exact algorithm. For graph pairs with more than 10 nodes, ground-truth labels are infeasible to obtain, we follow the strategy described in [11] to generate 100 synthetic graphs for each training graph with more than 10 nodes. Specifically, for a given graph, we apply $\Delta$ random edit operations to it, where $\Delta$ is randomly distributed in $[1, 10]$ for graphs with more than 20 nodes, and is randomly distributed in $[1, 5]$ for graphs with no more than 20 nodes. $\Delta$ is used as an approximation of the ground-truth GED.

To form the validation/testing graph pairs, each validation/testing graph with no more than 10 nodes is paired with 100 randomly selected training graphs with no more than 10 nodes. And each validation/testing graph with more than 10 nodes is paired with 100 synthetic graphs.

Table 3: Dataset statistics.

| Dataset | # Graphs | Avg $|V|$ | Avg $|E|$ | Max $|V|$ | Max $|E|$ | Number of Labels |
|---------|----------|-----------|-----------|-----------|-----------|------------------|
| AIDS700 | 700 | 8.9 | 8.8 | 10 | 14 | 29 |
| Linux | 1000 | 7.6 | 6.9 | 10 | 13 | 1 |
| IMDB | 1500 | 13 | 65.9 | 89 | 1467 | 1 |

## B.2 Implementation Details

**Network.** The denoising network $g_\phi$ consists of 6 layers with output dimensions $[128, 64, 32, 32, 32, 32]$, while the discriminator $D_\theta$ consists of 3 layers with output dimensions $[128, 64, 32]$.

**Training.** During training, the number of time steps $T$ for the forward process is set to 1000 with a linear noise schedule, where $\beta_0 = 10^{-4}$ and $\beta_T = 0.02$. For the Gumbel-Sinkhorn operator, the number of iterations $K$ is set to 5 and the temperature $\tau$ is set to 1. Moreover, we train $g_\phi$ and $D_\theta$ for 200 epochs with a batch size of 128. The loss weight $\lambda$ for $\mathcal{L}_{\text{explore}}$ in $\mathcal{L}_{g_\phi}$ (Equation 6) is linearly annealed from from 1 to 0 during the first 100 epochs, and fixed at 0 for the remaining 100 epochs. For the optimizer, We adopt RMSProp with a learning rate of 0.001 and a weight decay of $5 \times 10^{-4}$.

**Inference.** During inference, the number of time steps $S$ for the reverse process is set to 10 with a linear denoising schedule. For each test graph pair, we generate $k = 100$ candidate node matching matrices in parallel.

**Testbed.** All experiments are performed using an NVIDIA GeForce RTX 3090 24GB and an Intel Core i9-12900K CPU with 128GB RAM.

Table 4: Overall performance on unseen testing graph pairs.

| Datasets | Models | Type | MAE ↓ | Accuracy ↑ | $\rho$ ↑ | $\tau$ ↑ | p@10 ↑ | p@20 ↑ | Time(s) ↓ |
|---|---|---|---|---|---|---|---|---|---|
| | Hungarian | Trad | 8.237 | 1.5% | 0.527 | 0.416 | 54.3% | 60.3% | **0.0001** |
| | VJ | Trad | 14.171 | 0.9% | 0.391 | 0.302 | 44.9% | 52.9% | 0.00016 |
| | GEDGW | Trad | **0.828** | **53%** | **0.85** | **0.764** | **86.4%** | **85.8%** | 0.38911 |
| AIDS700 | Noah | SL | 3.174 | 6.8% | 0.735 | 0.617 | 77.8% | 76.4% | 0.5765 |
| | GENN-A* | SL | 0.508 | 67.1% | 0.917 | 0.836 | 87.1% | 90.6% | 3.44326 |
| | MATA* | SL | 0.885 | 56.6% | 0.77 | 0.689 | 73.2% | 76.6% | **0.00486** |
| | GEDGNN | SL | 1.155 | 50.5% | 0.838 | 0.746 | 89.1% | 87.6% | 0.39344 |
| | GEDIOT | SL | 1.348 | 47.4% | 0.81 | 0.71 | 88.4% | 86.9% | 0.39707 |
| | DiffGED | SL | **0.024** | **96.4%** | **0.993** | **0.986** | **99.7%** | **99.7%** | 0.07546 |
| | GEDRanker (Ours) | UL | **0.101** | **91.4%** | **0.981** | **0.963** | **99%** | **99%** | **0.07616** |
| | Hungarian | Trad | 5.423 | 7.5% | 0.725 | 0.623 | 75% | 77% | **0.00008** |
| | VJ | Trad | 11.174 | 0.4% | 0.613 | 0.512 | 70.6% | 74.5% | 0.00013 |
| | GEDGW | Trad | **0.568** | **73.5%** | **0.925** | **0.865** | **90.9%** | **91.5%** | 0.17768 |
| Linux | Noah | SL | 1.879 | 8% | 0.872 | 0.796 | 84.3% | 92.2% | 0.25712 |
| | GENN-A* | SL | 0.142 | 92.9% | 0.976 | 0.94 | 99.6% | 99.6% | 1.17702 |
| | MATA* | SL | 0.201 | 91.5% | 0.948 | 0.903 | 86.2% | 90.2% | **0.00464** |
| | GEDGNN | SL | 0.105 | 96.2% | 0.979 | 0.968 | 98.6% | 98.5% | 0.12169 |
| | GEDIOT | SL | 0.14 | 94.8% | 0.973 | 0.959 | 98.1% | 98.3% | 0.12826 |
| | DiffGED | SL | **0.0** | **100%** | **1.0** | **1.0** | **100%** | **100%** | 0.06901 |
| | GEDRanker (Ours) | UL | **0.008** | **99.6%** | **0.998** | **0.997** | **99.5%** | **99.7%** | **0.07065** |
| | Hungarian | Trad | 21.156 | 45.9% | 0.776 | 0.717 | 84.2% | 82.1% | **0.00012** |
| | VJ | Trad | 44.072 | 26.6% | 0.4 | 0.359 | 60.1% | 63.1% | 0.00037 |
| | GEDGW | Trad | 0.349 | 93.9% | 0.966 | 0.953 | 99.2% | 98.3% | 0.37384 |
| IMDB | Noah | SL | - | - | - | - | - | - | - |
| | GENN-A* | SL | - | - | - | - | - | - | - |
| | MATA* | SL | - | - | - | - | - | - | - |
| | GEDGNN | SL | 2.484 | 85.5% | 0.895 | 0.876 | 92.3% | 91.7% | 0.42662 |
| | GEDIOT | SL | 2.83 | 84.4% | 0.989 | 0.876 | 92.5% | 92.4% | 0.42269 |
| | DiffGED | SL | **0.932** | **94.6%** | **0.982** | **0.974** | **97.5%** | **98.4%** | **0.15107** |
| | GEDRanker (Ours) | UL | **1.025** | **93.9%** | **0.998** | **0.968** | **96%** | **97%** | **0.15012** |

# C More Experimental Results

## C.1 Generalization Ability

To evaluate the generalization ability of GEDRanker, we construct testing graph pairs by pairing each testing graph with 100 unseen testing graphs, instead of with training graphs. The overall performance on these unseen pairs is reported in Table 4. On unseen testing pairs, our unsupervised GEDRanker continues to achieve near-optimal performance, with only a marginal gap compared to the supervised DiffGED, and consistently outperforms all other supervised and traditional baselines. Furthermore, compared to the results in Table 1, there is no significant performance drop, demonstrating the generalization ability of our GEDRanker.

## C.2 Scalability

Table 5: Overall performance on large testing graph pairs.

| Datasets | Models | Average GED ↓ | Time(s) ↓ |
|---|---|---|---|
| | Hungarian | 234.656 | **0.00016** |
| | VJ | 219.465 | 0.00039 |
| IMDB-large | GEDGW | **140.278** | 0.4032 |
| | DiffGED (small) | 143.607 | 0.1911 |
| | GEDRanker (Ours) | **140.138** | **0.1904** |

Table 6: Ablation study on $D_\theta$'s network architecture.

| Datasets | Models | MAE ↓ | Accuracy ↑ | $\rho$ ↑ | $\tau$ ↑ | $p@10$ ↑ | $p@20$ ↑ |
|---|---|---|---|---|---|---|---|
| AIDS700 | GEDRanker (cost) | 0.528 | 66.3% | 0.907 | 0.841 | 90.4% | 92% |
| | GEDRanker | **0.088** | **92.6%** | **0.984** | **0.969** | **99.1%** | **99.1%** |
| Linux | GEDRanker (cost) | 0.102 | 94.9% | 0.98 | 0.965 | 96.8% | 97.1% |
| | GEDRanker | **0.01** | **99.5%** | **0.997** | **0.995** | **100%** | **99.8%** |

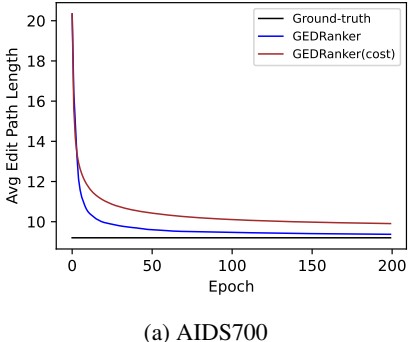

(a) AIDS700

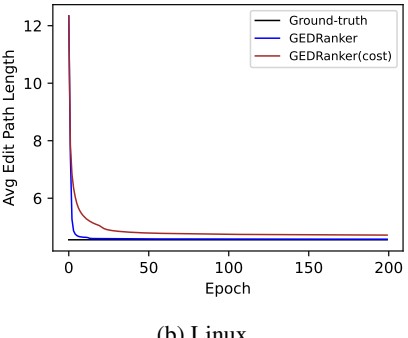

(b) Linux

Figure 6: Impact of $D_\theta$'s network architecture on the average edit path length of the best found node matching matrices on training graph pairs.

To evaluate the scalability of GEDRanker, we construct training pairs by pairing each large training graph in the IMDB dataset with all other large training graphs, instead of using synthetic graphs. For testing, we form pairs by pairing each large testing graph with 100 large training graphs. Since the scalability of GEDRanker can be inherently influenced by the base model, DiffGED. Therefore, an appropriate way to demonstrate our method's effectiveness on large graphs is to evaluate how it enhances the scalability of the base model in practice. To do so, we train the base model (DiffGED) only on small real-world IMDB graph pairs where ground-truth labels are available, then evaluate on the large testing graph pairs to see the scalability of the original base model, and compare with our GEDRanker that enables the training of base model on large graph pairs without requiring ground-truth supervision. For a clear reference, we also compare with traditional methods, and we report the average estimated GED across all testing pairs, along with the average inference time per testing pair.

Table 5 summarizes the overall performance of each method. Under this practical setting, we can see that the supervised DiffGED underperforms the optimization algorithm GEDGW in terms of average GED. However, by leveraging our unsupervised training strategy (GEDRanker), DiffGED can be trained on large graphs and subsequently outperforms GEDGW in both average GED and running time, hightlighting its scalability.

### C.3 Ablation Study on the Network Architecture of $D_\theta$

To evaluate the impact of $D_\theta$'s network architecture on the performance of GEDRanker, we create a variant model, *GEDRanker (cost)*, that adopts an alternative network design inspired by typical architectures used in regression-based GED estimation [11, 12]. Figure 5(b) shows an overview of the network architecture of $D_\theta$ in GEDRanker (cost).

Specifically, given a graph pair $(G_1, G_2)$ and a node matching matrix $\pi$, we compute nodes embeddings using GIN. Based on the pairwise node interactions, we then construct a matching cost matrix $C \in \mathbb{R}^{|V_1| \times |V_2|}$. Finally, the score $D_\theta(G_1, G_2, \pi)$ is estimated by $\langle \pi, C \rangle$. The overall architecture of $D_\theta$ can be represented as follows:

Table 7: Ablation study on loss weight $\lambda$.

| Datasets | Models | MAE $\downarrow$ | Accuracy $\uparrow$ | $\rho\uparrow$ | $\tau\uparrow$ | $p@10\uparrow$ | $p@20\uparrow$ |
|---|---|---|---|---|---|---|---|
| AIDS700 | GEDRanker (plain) | 0.549 | 65.4% | 0.905 | 0.837 | 91.7% | 91.5% |
| | GEDRanker (explore) | 1.334 | 39.3% | 0.789 | 0.689 | 76.5% | 79.9% |
| | GEDRanker | **0.088** | **92.6%** | **0.984** | **0.969** | **99.1%** | **99.1%** |
| Linux | GEDRanker (plain) | 0.079 | 96.2% | 0.984 | 0.973 | 98.1% | 98.3% |
| | GEDRanker (explore) | 0.146 | 93.4% | 0.973 | 0.955 | 96.6% | 98.2% |
| | GEDRanker | **0.01** | **99.5%** | **0.997** | **0.995** | **100%** | **99.8%** |

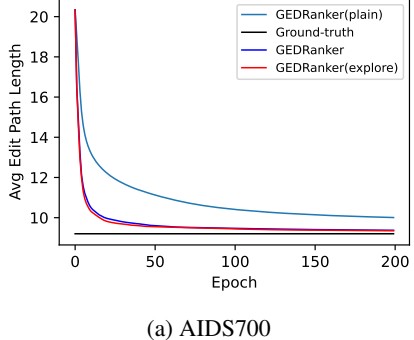

(a) AIDS700

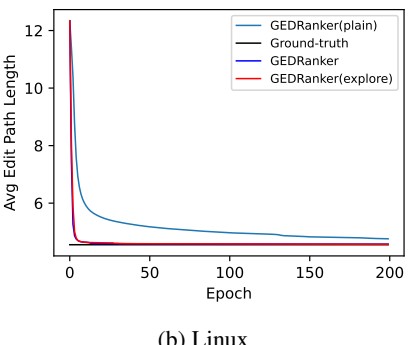

(b) Linux

Figure 7: Impact of $\lambda$'s network architecture on the average edit path length of the best found node matching matrices on training graph pairs.

$$
\begin{aligned}
\boldsymbol{h}_v^l &= \mathrm{GN}_{G_1}(\mathrm{MLP}((1+\epsilon^l)\cdot\boldsymbol{h}_v^{l-1} + \sum_{v'\in\mathcal{N}_{G_1}(v)}\boldsymbol{h}_{v'}^{l-1})) \\
\boldsymbol{h}_u^l &= \mathrm{GN}_{G_2}(\mathrm{MLP}((1+\epsilon^l)\cdot\boldsymbol{h}_u^{l-1} + \sum_{u'\in\mathcal{N}_{G_2}(u)}\boldsymbol{h}_{u'}^{l-1})) \\
\boldsymbol{H}_1 &= [\boldsymbol{h}_v^L]_{v\in V_1}, \quad \boldsymbol{H}_2 = [\boldsymbol{h}_u^L]_{u\in V_2} \\
\boldsymbol{C} &= MLP([\boldsymbol{H}_1\boldsymbol{W}_1\boldsymbol{H}_2^\top, \boldsymbol{H}_1\boldsymbol{W}_2\boldsymbol{H}_2^\top, ..., \boldsymbol{H}_1\boldsymbol{W}_n\boldsymbol{H}_2^\top]) \\
D_\theta(G_1, G_2, \pi) &= \langle\pi, \boldsymbol{C}\rangle
\end{aligned}
\tag{16}
$$

where we set $L=3$ and $n=16$.

Table 6 shows the overall performance of GEDRanker (cost). Surprisingly, GEDRanker (cost) performs significantly worse than GEDRanker. This phenomenon can be attributed to the following reasons: (1) Although GEDRanker (cost) estimates the score by assigning each node pair in $\pi$ an individual cost from $\boldsymbol{C}$, these costs are fixed for each node pair, irrespective of the actual value in $\pi$. The final score is computed as a simple linear combination $\langle\pi, \boldsymbol{C}\rangle$, which neglects the complex dependencies and interactions among node pairs. In contrast, $D_\theta$ in our GEDRanker directly computes node pair embeddings conditioned on the value in $\pi$, and leverages AGNN to capture interactions among node pairs; (2) The node matching matrix $\pi$ is inherently sparse, with most of its elements being 0. This sparsity significantly limits the ability of $D_\theta$ to learn correct $\boldsymbol{C}$ as only a few nonzero elements contribute to gradient updates. In contrast, $D_\theta$ in our GEDRanker transforms each value in $\pi$ into embeddings that enables effective gradient updates even the value is 0 in $\pi$.

Consequently, $D_\theta$ in GEDRanker (cost) struggles to estimate the correct preference order over different $\pi$, thereby misguiding $g_\phi$'s exploration direction, as demonstrated in Figure 6. This ultimately results in inferior performance.

### C.4 Ablation Study on Loss Weight $\lambda$

In our GEDRanker, $g_\phi$ is trained to minimize $\mathcal{L}_{g_\phi} = \mathcal{L}_{rec(\bar{\pi})} + \lambda\mathcal{L}_{explore}$, where the loss weight $\lambda$ is dynamically annealed during training to prioritize exploration in the early stages and shift $g_\phi$'s

Table 8: Ablation study on the decay schedule of $\lambda$.

| Decay Schedule | Datasets | MAE $\downarrow$ | Accuracy $\uparrow$ | $\rho\uparrow$ | $\tau\uparrow$ | $p@10\uparrow$ | $p@20\uparrow$ |
|---|---|---|---|---|---|---|---|
| Linear | AIDS | **0.088** | **92.6%** | **0.984** | **0.969** | 99.1% | 99.1% |
|  | Linux | **0.01** | **99.5%** | **0.997** | **0.995** | **100%** | **99.8%** |
| Cosine | AIDS | 0.130 | 90.2% | 0.977 | 0.955 | **99.2%** | 98.9% |
|  | Linux | 0.012 | 99.4% | 0.996 | 0.994 | 99.8% | 99.7% |
| Step | AIDS | 0.095 | 91.7% | 0.981 | 0.963 | 99.0% | **99.2%** |
|  | Linux | 0.013 | 99.4% | 0.996 | 0.994 | 99.9% | 99.7% |

focus toward recovering high-quality (exploitation) in the later stages. In Section 5.3, we evaluated GEDRanker (plain), which trains $g_\phi$ without any exploration by setting $\lambda = 0$. To understand the impact of prioritizing exploration through the entire training process, we create a variant model, *GEDRanker (explore)*, where $\lambda$ is fixed at 1.

Table 7 compares the performance of GEDRanker under different settings of $\lambda$. It is unexpected to observe that GEDRanker (explore) performs extremely poorly on the AIDS700 dataset. Although GEDRanker (explore) demonstrates strong exploration ability comparable to GEDRanker, as illustrated in Figure 7, its performance is still significantly worse.

The reason for this phenomenon lies in the mismatch between the training objective and the reverse diffusion process during inference. In a standard diffusion model, the reverse process is designed to gradually remove noise from the noisy node matching matrix step by step through a Markov chain (see Equation 2). During this process, $g_\phi$ is expected to correctly remove the noise from $\pi^t$. However, GEDRanker (explore) prioritizes exploration throughout the entire training process. This excessive focus on exploration prevents $g_\phi$ from learning the essential noise-removal patterns required by the reverse process. Although $g_\phi$ might still output high-quality node matching probability matrices at some steps, it still disrupts the reverse diffusion path, causing significant misalignment with the expected denoising trajectory. Consequently, errors are accumulated during reverse process, leading to inferior performance.

In contrast, our GEDRanker dynamically decreases $\lambda$ during training, promoting strong exploration in the early stages, while gradually shifting focus toward effective denoising of $\pi^t$ in alignment with the reverse diffusion process. This balanced strategy enables both thorough exploration during training and high-quality solutions during inference.

## C.5   Ablation Study on the Decay Schedule of $\lambda$

For GEDRanker, we adopt a linear decay schedule for $\lambda$. To investigate the impact of different decay strategies, we further experiment with cosine decay and step decay: (1) Cosine decay provides a similarly smooth transition as linear decay. Compared with the linear scheduler, it assigns more weight to exploration during the early stages and reduces the exploration weight more aggressively in the later stages. (2) Step decay maintains $\lambda = 1$ throughout the entire exploration phase and then directly drops it to 0 for exploitation.

As shown in Table 8, the performance of GEDRanker under different decay schedules exhibits no significant gap, indicating that our method is effective and robust without relying on complex scheduling strategies.

## C.6   Compatibility

Our unsupervised training method is not limited to DiffGED and can be easily integrated with other hybrid matching-based frameworks. To demonstrate the compatibility of GEDRanker, we integrate it with GEDGNN. As shown in Table 9, GEDGNN trained with our unsupervised method achieves performance comparable to that of GEDGNN trained with supervision, highlighting both the effectiveness and compatibility of our approach.

Table 9: Performance of GEDGNN trained with our unsupervised method.

| Setting | Datasets | MAE ↓ | Accuracy ↑ | $\rho$ ↑ | $\tau$ ↑ | $p@10$ ↑ | $p@20$ ↑ |
|---|---|---|---|---|---|---|---|
| | AIDS | 1.098 | 52.5% | 0.845 | 0.752 | 89.1% | 88.3% |
| Supervised | Linux | 0.094 | 96.6% | 0.979 | 0.969 | 98.9% | 99.3% |
| | IMDB | 2.469 | 85.5% | 0.898 | 0.879 | 92.4% | 92.1% |
| | AIDS | 1.279 | 44.6% | 0.809 | 0.712 | 82.6% | 83.7% |
| Unsupervised | Linux | 0.078 | 96.3% | 0.982 | 0.971 | 98.2% | 98.7% |
| | IMDB | 2.536 | 85.2% | 0.896 | 0.876 | 92.0% | 91.9% |

# D    Discussion of Limitations

Although GEDRanker has demonstrated promising results on the three most commonly used real-world GED datasets, one notable limitation of this paper is that, for pairs of larger graphs, obtaining ground-truth GED values becomes infeasible. As a result, we can only evaluate the differences in estimated GED values among baseline methods, rather than the actual gap between the estimated GED values and the real GED values. This also prevents us from evaluating ranking-based metrics for each test query graph in such scenarios.

