# OpenReview forum: "Towards Unsupervised Training of Matching-based Graph Edit Distance Solver via Preference-aware GAN"
_NeurIPS.cc/2025/Conference — NeurIPS 2025 poster_

### Official Review · Reviewer_Y63G · 2025-06-29

**Clarity:** 2
**Significance:** 3
**Originality:** 3
**Rating:** 4
**Confidence:** 3

**Summary:**

The paper provides a novel method to approximate the graph edit distance (GED), by investigating an unsupervised framework thanks to a reformulation using a generative adversarial network (GAN).

**Questions:**

no further questions.

**Ethical Concerns:**

["NO or VERY MINOR ethics concerns only"]

**Final Justification:**

I thank the authors for the clarifications on all the major issues raised in my review, which were mainly related to the positioning of this work in the literature. Considering the overall reviews and the rebuttal discussions, I have accordingly raised my score.

**Limitations:**

yes

**Quality:**

2

**Strengths And Weaknesses:**

The major strength of this work is that it seeks to provide an unsupervised framework for learning the GED.

The paper suffers from several major weaknesses.

The authors pretend that this is the first unsupervised learning framework for GED computation, as given for instance in the subsection “Contributions” at the end of Section 1. However, this is not correct, since there are many other unsupervised learning methods for GED computation. See for instance:
- Bommakanti, A., Vonteri, H. R., Ranu, S., & Karras, P. (2024). EUGENE: Explainable Unsupervised Approximation of Graph Edit Distance. arXiv preprint arXiv:2402.05885.
That paper also provides several references to other works that provide unsupervised methods, such as F1 (Lerouge et al., 2017), ADJ-IP (Justice & Hero, 2006), COMPACT-MIP (Blumenthal & Gamper, 2020), and BRANCH-TIGHT (Blumenthal & Gamper, 2018).
Overall, the paper is not well positioned within the literature of related work, and thus the experiments are missing a comparative analysis with such related work.

The main contribution of the paper seems to be the usage of the Gumbel-Sinkhorn method, as opposed to REINFORCE. While this seems to be sound, the authors need to highlight this contribution within the appropriate literature. It seems that the authors are not aware of the broad work on the usage of the Gumbel-Sinkhorn method to compute the GED. See for instance:
- Jain, E., Roy, I., Meher, S., Chakrabarti, S., & De, A. (2024). Graph edit distance with general costs using neural set divergence. Advances in Neural Information Processing Systems, 37, 73399-73438.
- Wang, J., Zhu, H., Xie, H., Wang, F. L., Xu, X., & Wang, Y. (2024). Graph Similarity Computation via Interpretable Neural Node Alignment. arXiv preprint arXiv:2412.12185.
- Pei, M., Yu, J., Chen, C., Wang, H., Wang, X., & Zhang, Y. (2025). IGFM: An Enhanced Graph Similarity Computation Method with Fine-Grained Analysis. Data Science and Engineering, 1-15.
- Roy, I., Jain, E., Meher, S., Chakrabarti, S., & De, A. (2024). Graph Edit Distance with General Costs Using Neural Set Divergence. In The Third Learning on Graphs Conference. https://openreview.net/pdf?id=rpF6eqVkPu

The paper is difficult to read and understand because some parts are overstressed and others are not described enough.

It feels weird how the problem is presented using REINFORCE from a 1992 paper, before seeking to overcome the limitations of that work that is more than 30 years old.

The description of the supervised diffusion-based node matching model [13] in Section 3.2 is not justified explicitly. One can wonder about the interest of that model within the proposed method. Besides the caption of Figure 2 that provides little insight, the main text does not clearly describe the integration of that diffusion-based model.

At the end, the proposed method relies mainly on the work [13], which is a preprint in arXiv that seems not to be peer-reviewed (to the best of our knowledge).

On the quality of writing, there is room for improvement in order to make the paper more accessible to readers. For instance, GAN is never defined in the paper, although it is central to this work.

---

> ### Author Rebuttal · Authors · 2025-07-30
>
> We would like to clarify the following points.
>
> ---
>
> ### **W1. Clarification on EUGENE and optimization-based methods**
>
> EUGENE is actually an **optimization-based method** (similar in nature to GEDGW), it is **NOT a neural unsupervised learning method** (it does not involve any neural network/deep learning). Moreover, the so-called "unsupervised methods" referenced by EUGENE are also **optimization methods** (e.g., linear programming), **NOT neural unsupervised methods**.
>
> The key distinction between optimization and neural unsupervised methods lies in generalizability: **optimization methods do not involve neural networks**, and they **cannot be trained on training graphs**, instead, their **optimization process must be repeated for each test graph pair** (of course in this case, they could in some sense be called ''unsupervised''), making them non-generalizable and time consuming. In contrast, neural unsupervised methods, once trained, can be **directly applied to unseen test graph pairs via neural networks without further optimization**. And our framework is the first method that **enables the unsupervised training of supervised neural methods**. Moreover, we have already compared and outperformed the latest SOTA optimization algorithm, GEDGW, recently accepted by SIGMOD2025.
>
> ---
>
> ### **W2. Clarification on the main contribution**
>
> **Gumbel-Sinkhorn is NOT our main contribution**, we have clearly stated that **the main contribution of our work is the preference-aware GAN-based training framework** (line41-49). Gumbel-Sinkhorn only serves as a tool to enable smooth gradient flow between the generator and the discriminator.
>
> Also, please note that the objective of our task is to **find the actual edit path that minimizes the total cost** (discrete value), where the ground-truth labels are the edit paths/matching matrices in the supervised case. In contrast, for the papers you referenced, they focus on **formulating GED as a regression problem** (predicts a continuous value), where their ground-truth labels are the GED scores. The nature of the objectives/outputs is **completely different**.
>
> While regression-based approach may achieve very low MAE by making continuous number prediction, **their accuracy can be very poor and they can make infeasible predictions** (e.g., the predicted GED < the actual GED). Therefore, regression-based approach is **NOT the focus of our work**.
>
> ---
>
> ### **W3. Clarification on REINFORCE**
>
> Although REINFOCE is introduced in 1992, it is still the most well-known and widely used fundamental unsupervised RL method, particularly in lots of unsupervised frameworks for other NP-hard problems (even in LLM+RL). For example:
>
>     André Hottung, Yeong-Dae Kwon, Kevin Tierney: Efficient Active Search for Combinatorial Optimization Problems. ICLR 2022
>     Qiu, R., Sun, Z., & Yang, Y. (2022). Dimes: A differentiable meta solver for combinatorial optimization problems. Advances in Neural Information Processing Systems, 35, 25531-25546.
>     Ye, H., Wang, J., Liang, H., Cao, Z., Li, Y., & Li, F. (2024, March). Glop: Learning global partition and local construction for solving large-scale routing problems in real-time. In Proceedings of the AAAI Conference on Artificial Intelligence (Vol. 38, No. 18, pp. 20284-20292).
>     Back to Basics: Revisiting REINFORCE-Style Optimization for Learning from Human Feedback in LLMs (Ahmadian et al., ACL 2024).
>
> ---
>
> ### **W4. Clarification on the integration with the diffusion-based model**
>
> The majority of Section 4 focuses on replacing the supervised training strategy of the diffusion-based model described in Section 3.2 with our unsupervised training strategy (line 142-148). In particular, Section 4.1 and Algorithm 3 provide all the necessary details for integrating GEDRanker with the diffusion-based model (see line 3 of Algorithm 3). Figure 2 illustrates the complete unsupervised training pipeline, the only part it needs to be integrated with the diffusion-based model is the forward process $q(\pi^t|\pi^0)$.
>
> Also, please note that, our framework is not specifically designed for the diffusion-based model only. We select the diffusion-based model as our base model due to its superior performance and efficiency over all other existing hybrid methods, our framework can be easily adopted to future SOTA matching-based methods. To demonstrate this, we train an older matching-based approach GEDGNN with our unsupervised approach:
> | Setting      | Dataset | MAE   | Accuracy   | ρ     | τ     | pk10  | pk20  |
> |--------------|---------|-------|-------|-------|-------|-------|-------|
> | Supervised   | AIDS    | 1.098 | 52.5% | 0.845 | 0.752 | 89.1% | 88.3% |
> |              | LINUX   | 0.094 | 96.6% | 0.979 | 0.969 | 98.9% | 99.3% |
> |              | IMDB    | 2.469 | 85.5% | 0.898 | 0.879 | 92.4% | 92.1% |
> | Unsupervised | AIDS    | 1.279 | 44.6% | 0.809 | 0.712 | 82.6% | 83.7% |
> |              | LINUX   | 0.078 | 96.3% | 0.982 | 0.971 | 98.2% | 98.7% |
> |              | IMDB    | 2.536 | 85.2% | 0.896 | 0.876 | 92.0% | 91.9% |
>
> The performance of GEDGNN trained with our unsupervised method is similar to that of GEDGNN trained under supervision, which demonstrates the compatibility and the effectiveness of our apporach.
>
> ---
>
> ### **W5. Clarification on DiffGED**
>
> We have provided the implementations of all baseline methods including DiffGED. Through our implementation and experimental results, we demonstrate that DiffGED is indeed highly effective and efficient, which justifies our choice of it as the base model.
>
> ---
>
> ### **W6. Clarification on GAN**
>
> GAN is a well-known framework. We have already provided an overview of its core idea in the context of our work in Section 4.1. (GAN-based Training Objective line180-184)

---

> > ### Comment · Reviewer_Y63G · 2025-08-07
> >
> > I thank the authors for the clarifications on all the major issues raised in my review, which were mainly related to the positioning of this work in the literature. Considering the overall reviews and the rebuttal discussions, I have accordingly raised my score.

---

### Official Review · Reviewer_9zJG · 2025-07-01

**Clarity:** 4
**Significance:** 3
**Originality:** 3
**Rating:** 5
**Confidence:** 4

**Summary:**

Recent neural frameworks have been shown to be effective at learning the Graph Edit Distance (GED). However, these supervised frameworks require ground-truth GED for training, which is NP-hard to compute. The paper proposes an unsupervised technique to overcome this issue. The backbone model is diffusion-based and trained to generate a node mapping matrix, from which an edit path can be generated. Instead of matching the ground-truth node mapping matrix (which is unavailable), the proposed method tries to match the best mapping matrix found so far, which is iteratively updated after being compared with the new matrix returned by the model. In addition, the method tries to reduce the resulted edit distance directly. In order to do this, the authors introduce a discriminator $D$ which takes as input the two graphs and a mapping matrix and output a score reversely proportional to the edit distance. The additional objective is to maximize $D(G_1, G_2, \pi)$ where $\pi$ is the mapping matrix returned by the model (and passed through a Gumbel-Sinkhorn operator for differentiability). The discriminator uses the Bayes Personalized Ranking loss to better enforce the ordering of the edit paths, i.e. a longer edit path will have a lower score. Overall, I have enjoyed reading this paper.

**Questions:**

This paper is extremely well-written and I enjoyed reading it! Only major concern is W1 in weaknesses. A few more questions are as follows:

- In Equation 4, how is the cost function $c(\cdot)$ calculated? Is it based on the procedure of Edit Path Generation written in section 3.1? This should be specified in the paper.
- How compatible is your method with other hybrid approaches besides DiffGED?
- In section C.1 in the appendix, how well is your method performing on \textbf{large}, unseen graph pairs, in comparison to other baselines? In other words, how is generalizability of the proposed method affected by the graph size?
- Have you tried other scheduler for the dynamical decrease of $\lambda$ other than the linear scheduler?
- A few typos: sapce (line 247),  categorizebaseline (line 255).

**Ethical Concerns:**

["NO or VERY MINOR ethics concerns only"]

**Final Justification:**

The rebuttal has addressed the comments with new experiments.

**Limitations:**

They have adequately addressed it.

**Quality:**

3

**Strengths And Weaknesses:**

**Strengths:**

- The paper touches upon an important and relatively underexplored problem: supervised methods for learning GED require ground-truth labels that are NP-hard to compute.

- The paper proposes a novel solution: encouraging the model to progressively come up with better answers. In addition, the length of the edit path is reduced directly via a clever use of the Discriminator and the Gumbel-Sinkhorn trick to get around the indifferentiability caused by working in the discrete graph space.

- A wide range of evaluation metrics are used and reported.

**Weaknesses:**

- W1: Datasets of larger graphs (e.g. the OGB datasets) should be used for evaluation. This is because the main motivation for the paper is the NP-hardness of computing the ground-truth GED, which becomes costly for larger graphs. For IMDB-Large, the method returns a very high edit path (in absolute terms), which is undesirable. This is acknowledged in the Limitations section.

- W2: The regression-based neural frameworks (SimGNN and its successors) should also be included as baselines. There can be applications in which only the numerical GED value is important.

---

> ### Author Rebuttal · Authors · 2025-07-30
>
> Thanks for enjoying reading our paper XD! Below we address each point in detail.
>
> ---
>
> ### **W1. Scalability**
>
> We would like to clarify that, for IMDB-large dataset, the founded average GED = 140.138 is **NOT necessarly high** for large graphs (e.g., the range of GED for a pair of graphs with 50 nodes is [0,50*49/2+50=1275]), we just **don't know how far it is from the actual optimal GED**. But comparing with the traditional methods, our method can find **a smaller GED and with a shorter time**. And IMDB-large is sufficiently large for GED computation compared to other widely-used GED datasets, as traditional exact algorithms could take $10^2$ seconds even just on a single graph pair of 30 nodes.
>
> Please also note that the scalability can be **inherently influenced by the base model**, DiffGED. Therefore, we think a more appropriate way to demonstrate our method's effectiveness on large graphs is by evaluating **how it enhances the scalability of the base model in practice**. To this end, we train the base model (DiffGED) only on small real-world IMDB graph pairs where ground-truth labels are available. We then evaluate it on the IMDB-large graph pairs used in Table 5 to see the scalability of the original base model, and compared with our GEDRanker that enables the training of base model on large graph pairs without requiring ground-truth supervision. Following are the evaluation results:
>
> | Method     | Average GED | Time(s) |
> |------------|-------------|---------|
> | DiffGED    | 143.607      | 0.1911  |
> | GEDGW      | 140.278      | 0.4032  |
> | GEDRanker  | 140.138      | 0.1904  |
>
> Under this practical setting, we can see that the supervised DiffGED underperforms the optimization algorithm GEDGW in terms of average GED. However, by leveraging our unsupervised training strategy (GEDRanker), DiffGED can be trained on large graphs and subsequently outperforms GEDGW in both average GED and running time. It is worth noting that GEDGW is an **optimization algorithm**, meaning its performance cannot be further improved easily **without extra time cost on the test instances** (and some of them can only output a fixed approximation). In contrast, neural methods **offer the potential for performance improvement** through increased training data, longer training epochs, or other techniques, these do not require extra time cost on the test instances. Therefore, comparison with the optimization algorithms is **NOT the primary focus** of this work.
>
> ---
>
> ### **W2. Regression-based baselines**
>
> Regression-based neural frameworks are **NOT the focus of our work**, as **their objective is NOT to minimize the edit cost** (their ground-truth labels are the GED scores and they make continuous value prediction). While they may achieve very low MAE by making continuous number prediction, their accuracy can be very poor and they can make **infeasible predictions** (e.g., the predicted GED < the actual GED). But we can include them as interesting baselines.
>
> ---
>
> ### **Q1. Cost function**
>
> Yes, the cost function is computed based on the lenght of the founded edit path, we will specify it in the paper.
>
> ---
>
> ### **Q2. Integration with other hybrid approaches**
>
> We choose DiffGED as our base model, as it significantly outperforms all other hybrid approaches in terms of both effectiveness and efficiency. However, our method is **NOT limited to DiffGED** and can be **easily integrated with other hybrid matching-based frameworks**. For example, the following results show the performance of GEDGNN when trained using our unsupervised method:
>
> | Setting      | Dataset | MAE   | Accuracy   | ρ     | τ     | pk10  | pk20  |
> |--------------|---------|-------|-------|-------|-------|-------|-------|
> | Supervised   | AIDS    | 1.098 | 52.5% | 0.845 | 0.752 | 89.1% | 88.3% |
> |              | LINUX   | 0.094 | 96.6% | 0.979 | 0.969 | 98.9% | 99.3% |
> |              | IMDB    | 2.469 | 85.5% | 0.898 | 0.879 | 92.4% | 92.1% |
> | Unsupervised | AIDS    | 1.279 | 44.6% | 0.809 | 0.712 | 82.6% | 83.7% |
> |              | LINUX   | 0.078 | 96.3% | 0.982 | 0.971 | 98.2% | 98.7% |
> |              | IMDB    | 2.536 | 85.2% | 0.896 | 0.876 | 92.0% | 91.9% |
>
> The performance of GEDGNN trained with our unsupervised method is similar to that of GEDGNN trained under supervision, which demonstrates the effectiveness of our apporach.
>
> Here we don't need to show the performance of GEDIOT trained with our unsupervised framework, this is because GEDIOT adopts the same architecture as GEDGNN, with the only difference being an additional Sinkhorn layer applied to the output node matching matrix. To adapt GEDIOT to our unsupervised training framework, we simply replace the Sinkhorn layer with the Gumbel-Sinkhorn trick, this naturally becomes GEDGNN + our unsupervised method.
>
> ---
>
> ### **Q3. Generalization ability w.r.t. graph size**
>
> To better demonstrate the generalizability of our method **with respect to the graph size**, we follow the same setting as described in the original paper of DiffGED. Specifically, instead of training each method on a combination of real small graph pairs and synthetic large graph pairs from IMDB, we train each method on real small graph pairs only, then evaluate on the whole IMDB dataset that contains a combination of small real graph pairs and large synthetic graph pairs:
>
> | Method     | MAE   | Accuracy   | ρ     | τ     | pk10  | pk20  |
> |------------|-------|-------|-------|-------|-------|-------|
> | GEDGNN     | 7.943 | 77.1% | 0.844 | 0.815 | 88.2% | 87.6% |
> | GEDIOT     | 7.761 | 76.8% | 0.860 | 0.827 | 90.5% | 89.9% |
> | DiffGED    | 5.789 | 83.0% | 0.892 | 0.874 | 90.1% | 90.8% |
> | GEDRanker  | 8.564 | 81.0% | 0.860 | 0.844 | 87.6% | 88.3% |
>
> Please be aware of that, these results are evaluated based on the **approximated** synethic ground-truth (might be larger than the exact ground-truth), therefore they only indicate the **distance between the predicted GED and the approximated ground-truth GED** (the predicted GED could be better than the approximated ground-truth GED).
>
> Moreover, the overall generalization ability is not only affected by our unsupervised training method, but can also be **affected by the generalization ability of the base model**.
>
> Additionally, our GEDRanker **faces a more challenging setting** compared to baseline methods: our GEDRanker **not only need to generalize** to large unseen graphs, but also need to be **trained without any ground-truth supervision on small graphs**, where baseline models can be trained with ground-truth supervision on small graphs. Under this more challenging setting, GEDRanker can still achieve similar generalization ability as DiffGED, which further demonstrates the effectiveness of our approach.
>
> ---
>
> ### **Q4. Evaluation of $\lambda$ schedule**
>
> First, we would like to clarify that the primary focus of our work is to **efficiently explore high-quality solutions within a limited number of epochs**. If we can efficiently explore high quality solutions during the exploration stage, then we are able to shift toward exploitation using a **simple controllable decay**, **without the need** for designing complex exploration-exploitation balance mechanism and tunning.
>
> To demonstrate this, we also experimented with cosine decay and step decay: (1) Cosine decay provides a similar smooth transition as linear decay. Compared to the linear scheduler, cosine decay allocates more weight to exploration during the early exploration stages, and reduces the exploration weight more aggressively in the later exploration stages. (2) Step decay keeps $\lambda=1$ during the whole exploration phase then directly drops $\lambda$ to 0 for exploitation. Below are the results of GEDRanker trained with cosine decay and step decay:
>
> | Decay Type   | Dataset | MAE   | Accuracy    | ρ     | τ     | pk10  | pk20  |
> |--------------|---------|-------|--------|-------|-------|-------|-------|
> | Linear       | AIDS    | 0.088 | 92.6%  | 0.984 | 0.969 | 99.1% | 99.1% |
> |              | Linux   | 0.010 | 99.5%  | 0.997 | 0.995 | 100%  | 99.8% |
> |              | IMDB    | 1.109 | 94%    | 0.999 | 0.97  | 96.1% | 97%   |
> | Cosine       | AIDS    | 0.130 | 90.2%  | 0.977 | 0.955 | 99.2% | 98.9% |
> |              | Linux   | 0.012 | 99.4%  | 0.996 | 0.994 | 99.8% | 99.7% |
> |              | IMDB    | 1.037 | 94.5%  | 0.979 | 0.97  | 97.2% | 97.8% |
> | Step         | AIDS    | 0.095 | 91.7%  | 0.981 | 0.963 | 99.0% | 99.2% |
> |              | Linux   | 0.013 | 99.4%  | 0.996 | 0.994 | 99.9% | 99.7% |
> |              | IMDB    | 1.187 | 93.4%  | 0.97  | 0.96  | 96.7% | 97.4% |
>
> The results show that there's no significant performance gap between these 3 types of decay, this further demonstrates that our method is effective and robust, it does not require complex schedule design.

---

### Official Review · Reviewer_RQNJ · 2025-07-02

**Clarity:** 3
**Significance:** 3
**Originality:** 3
**Rating:** 5
**Confidence:** 3

**Summary:**

The authors propose GEDRanker, a novel unsupervised learning framework employing a GAN paradigm. GEDRanker consists of a matching-based GED solver, a preference-aware discriminator that provides a more interpretable and effective learning objective by evaluating and ranking node matching matrices based on their relative quality rather than relying on ground-truth supervision, and an effective training strategy that adaptively integrates exploitation and exploration during training.

In terms of performance, GEDRanker achieves near-optimal GED solution quality without supervision, consistently outperforming traditional approximate methods and closely matching supervised state-of-the-art methods in terms of accuracy and efficiency.

**Questions:**

1. The proposed method uses a preference-aware discriminator trained with a ranking loss. How stable is this training approach in practice, and how sensitive is the model's performance to the initialization of the discriminator? Were multiple runs with different seeds performed to evaluate training stability?

2. Have you explored or considered alternative exploration-exploitation strategies beyond linearly annealing $\lambda$ in your training objective? Not sure whether we should investigate adaptive strategies, e.g., reinforcement-learning-inspired adaptive exploration scheduling or dynamic $\lambda$ adjustments based on current performance trends.

3. How well does the discriminator generalize the learned preferences to completely unseen graph pairs that may differ structurally or in size distributions from those seen during training?

**Ethical Concerns:**

["NO or VERY MINOR ethics concerns only"]

**Final Justification:**

Overall, the authors have addressed all major concerns with strong empirical results and thoughtful reasoning. The proposed method is novel, effective, and well-supported by the experimental results. I recommend accept.

**Limitations:**

The authors have not explicitly addressed the limitations and potential negative societal impacts. The author should clearly discuss the known limitations regarding scalability or conditions under which GEDRanker may perform suboptimally.

**Quality:**

3

**Strengths And Weaknesses:**

Strengths:
1. The paper proposes the first unsupervised GAN-based approach for GED computation. Its unique use of a preference-aware discriminator provides an innovative direction compared to existing reinforcement-learning or supervised strategies.
2. The presentation is clear and it has comprehensive motivation and background effectively position the paper within existing literature.
3. The method demonstrates performance competitive with state-of-the-art supervised methods, emphasizing the value of unsupervised training and has strong experimental validation across widely-used benchmark datasets.


Weaknesses:
1. Hyperparameter sensitivity or case studies could be expanded to better demonstrate robustness and generalizability.
2. Practical significance could be even better supported by demonstrating the method’s performance or limitations explicitly in large-scale or more varied real-world scenarios.

---

> ### Author Rebuttal · Authors · 2025-07-30
>
> Thanks for providing constructive comments. Below we address each point in detail.
>
> ---
>
> ### **Q1. Stability**
>
> We would like to demonstrate the stability of our approach from several perspectives:
>
> - First, we have demonstrated the curve of the average edit path length of the best found solution during training in Figure 3, where we evaluated our GEDRanker with **an alternative ranking loss**, and we have discussed its stability in Section 5.3 (Ranking Loss). Overall, our proposed method is stable, and even with an alternative ranking loss, it can still achieve excellent performance.
>
> - To further demonstrate the stability of our approach, we train another supervised hybrid approach GEDGNN with our unsupervised approach:
>
>   >| Setting      | Dataset | MAE   | Accuracy   | ρ     | τ     | pk10  | pk20  |
>   >|--------------|---------|-------|-------|-------|-------|-------|-------|
>   >| Supervised   | AIDS    | 1.098 | 52.5% | 0.845 | 0.752 | 89.1% | 88.3% |
>   >|              | LINUX   | 0.094 | 96.6% | 0.979 | 0.969 | 98.9% | 99.3% |
>   >|              | IMDB    | 2.469 | 85.5% | 0.898 | 0.879 | 92.4% | 92.1% |
>   >| Unsupervised | AIDS    | 1.279 | 44.6% | 0.809 | 0.712 | 82.6% | 83.7% |
>   >|              | LINUX   | 0.078 | 96.3% | 0.982 | 0.971 | 98.2% | 98.7% |
>   >|              | IMDB    | 2.536 | 85.2% | 0.896 | 0.876 | 92.0% | 91.9% |
>
>   The performance of GEDGNN trained with our unsupervised method is similar to that of GEDGNN trained under supervision, this again demonstrates the robustness of our approach. Here we don't need to show the performance of GEDIOT trained with our unsupervised framework, this is because GEDIOT adopts the same architecture as GEDGNN, with the only difference being an additional Sinkhorn layer applied to the output node matching matrix. To adapt GEDIOT to our unsupervised training framework, we simply replace the Sinkhorn layer with the Gumbel-Sinkhorn trick, this naturally becomes GEDGNN + our unsupervised method.
>
> - Regarding the initialization of the discriminator, for the results reported in the paper, we used the same seed as baseline methods **for a fair comparsion**. To demonstrate the stability, we randomly initialize the discriminator over 4 seeds, following are the results with mean&std:
> >| Dataset | MAE (±)       | ACC (±)         | ρ (±)          | τ (±)          | pk10 (±)        | pk20 (±)        |
> >|---------|---------------|-----------------|----------------|----------------|-----------------|-----------------|
> >| AIDS    | 0.089 ± 0.0026 | 92.39% ± 0.43%  | 0.984 ± 0.003  | 0.968 ± 0.0026 | 99.07% ± 0.064% | 99.11% ± 0.028% |
> >| Linux   | 0.010 ± 0.0015 | 99.49% ± 0.0173%| 0.9966 ± 0.0006| 0.9948 ± 0.0005| 99.98% ± 0.05%  | 99.79% ± 0.017% |
> >| IMDB    | 1.016 ± 0.0064 | 94.08% ± 0.2217%  | 0.996 ± 0.0022 | 0.969 ± 0.0010 | 96.15% ± 0.2646%  | 97.08% ± 0.2217%  |
>
>   Overall, the performance is stable.
>
> - Please also see the response to Q2, which can further demonstrate the stability of our approach.
>
> ---
>
> ### **Q2. Evaluation of $\lambda$ schedule**
>
> - First, we would like to clarify that the primary focus of our work is to **efficiently explore high-quality solutions within a limited number of epochs**. If we can efficiently explore high quality solutions during the exploration stage, then we are able to shift toward exploitation using **a simple controllable decay**, **without the need for designing complex** exploration-exploitation balance mechanism and tunning.
>
> - To demonstrate this, we also experimented with cosine decay and step decay: (1) Cosine decay provides a similar smooth transition as linear decay. Compared to the linear scheduler, cosine decay allocates more weight to exploration during the early exploration stages, and reduces the exploration weight more aggressively in the later exploration stages. (2) Step decay keeps $\lambda=1$ during the whole exploration phase then directly drops $\lambda$ to 0 for exploitation. Below are the results of GEDRanker trained with cosine decay and step decay:
>
>   >| Decay Type   | Dataset | MAE   | Accuracy    | ρ     | τ     | pk10  | pk20  |
>   >|--------------|---------|-------|--------|-------|-------|-------|-------|
>   >| Linear       | AIDS    | 0.088 | 92.6%  | 0.984 | 0.969 | 99.1% | 99.1% |
>   >|              | Linux   | 0.010 | 99.5%  | 0.997 | 0.995 | 100%  | 99.8% |
>   >|              | IMDB    | 1.109 | 94%    | 0.999 | 0.97  | 96.1% | 97%   |
>   >| Cosine       | AIDS    | 0.130 | 90.2%  | 0.977 | 0.955 | 99.2% | 98.9% |
>   >|              | Linux   | 0.012 | 99.4%  | 0.996 | 0.994 | 99.8% | 99.7% |
>   >|              | IMDB    | 1.037 | 94.5%  | 0.979 | 0.97  | 97.2% | 97.8% |
>   >| Step         | AIDS    | 0.095 | 91.7%  | 0.981 | 0.963 | 99.0% | 99.2% |
>   >|              | Linux   | 0.013 | 99.4%  | 0.996 | 0.994 | 99.9% | 99.7% |
>   >|              | IMDB    | 1.187 | 93.4%  | 0.97  | 0.96  | 96.7% | 97.4% |
>
>   The results show that there's no significant performance gap between these 3 types of decay, this further demonstrates that our method is effective and robust, it does not require complex schedule design.
>
> - Regarding the RL-based adaptive exploration scheduling, it is not well-suited to our framework, as it can lead to **unstable fluctuations between exploration and exploitation**. Such instability may hinder the diffusion-based model from learning the correct denoising process, ultimately **breaking the denoising chain** during inference. Consequently, issues similar to those observed in GEDRanker (explore) in Appendix C.4 may arise (can find good solutions for training data, but performs poorly on unseen data). This would require a more sophisticated RL strategy specifically tailored to diffusion-based models, which is undesirable because our method is intended to **remain simple and easily adaptable to other supervised matching-based approaches** (this was demonstrated in the response to Q1). Our results demonstrate that a simple linear decay schedule is sufficient to achieve good performance.
>
> - Regarding the dynamic adjustments based on current performance trends, we have tried it, but found it **difficult to control**. For instance, when the model is able to discover only a few better solutions, it is unclear whether the exploration ratio should be increased or decreased. Reducing exploration may **trap the model in a locally sub-optimal region**, whereas increasing it may be also **harmful when the model does not need further exploration**. This again can **cause fluctuation between unstable exploration and exploitation**, and requires complex design and tunning.
>
> ---
>
> ### **Q3. Evaluation of the discriminator**
>
> Note that the discriminator is used **solely to guide the training of the matching model** and is **trained in an online manner** based on the distribution of latest solutions sampled from the matching model during exploration phase. It is **NOT used during inference**; therefore, it is **NOT expected to generalize to unseen graph pairs**. Moreover, the discriminator is **NOT trained to perfectly compute** the preference over all node matching matrices, it **only aims to provide a relatively good training signal** that guides the matching model within the current training epoch. Similar to those discriminators used in image generation, they are not intended to serve as a useful or generalizable image classifier after training.
>
> Therefore, we only need to evaluate **discriminator's online performance**, and the only thing we really need to care about is **whether the discriminator can help the generator to explore better solutions more efficiently during training**. This is demonstrated in Figure 3 of our paper, the discriminator is very effective in helping the generator to find near-optimal solutions for training data.
>
> Another evaluation of the discriminator can be showing **the loss curve of the discriminator during training**, but we are **not allowed to post the any visualization results** during the rebuttal period, we will show it in the camera-ready version.

---

> > ### Comment · Reviewer_RQNJ · 2025-08-05
> >
> > Thank you for the detailed explanation. I will keep my current score as accept.

---

### Official Review · Reviewer_eHfB · 2025-07-02

**Clarity:** 3
**Significance:** 3
**Originality:** 3
**Rating:** 5
**Confidence:** 2

**Summary:**

This paper proposes an unsupervised diffusion model training method for node matching between input two graphs. Instead of using supervised node matching information, the author trains the model so as to (i) mimic the current best coupling, (ii) reduce the cost of coupling (iii) increase the preference score obtained from discriminator. The proposed method demonstrates comparable accuracy to existing supervised methods. The author further conducts ablation study about the proposed modules.

**Questions:**

More evaluation on large scale datasets.

Evaluation on training cost.

Evaluation on the learned discriminator.

**Ethical Concerns:**

["NO or VERY MINOR ethics concerns only"]

**Final Justification:**

The author rebuttal addressed  my concerns. Therefore, I increase the score.

**Limitations:**

Discussed in the appendix.

**Paper Formatting Concerns:**

I could not find.

**Quality:**

3

**Strengths And Weaknesses:**

Strength

The author proposes novel task to train node matching unsupervisedly. To this end, the author proposes several training strategy and loss functions.

The proposed model learned unsupervisedly shows comparable performance to DiffGED trained supervisedly. This shows that this unsupervised approach is promising.

The author conducts several ablation study about the proposed modules. All the proposed modules contributes to the performance.


Weakness

I want to see the evaluation on the correlation between the discriminator’s preference and true edit distance.

I also want to see the evaluation on the training cost. Since we need to train additional discriminator module, I think we need to train longer epoch than supervised case.
Further, I am curious about the semi-supervised performance.

I think the performance may depend on the initial matching policy $\bar{\pi}$. What if we initialize $\bar{\pi}$ with current best unsupervised method such as GEDGW?

Also, I think the proposed method is effective in the larger-scale case since we cannot obtain the ground truth supervised information. However, in Table 5, the accuracy of the proposed method is close to the traditional GEDGW. I want to know more evaluation on this large-scale area.

---

> ### Author Rebuttal · Authors · 2025-07-30
>
> Thanks for providing constructive comments. Below we address each point in detail.
>
> ### **1. Evaluation of the discriminator**
>
> We first would like to clarify that the discriminator is used **solely to guide the training of the matching model** and is **trained in an online manner** based on the distribution of latest solutions sampled from the matching model during exploration phase. It is **NOT used during inference**; therefore, it is **NOT trained to perfectly compute** the preference over all node matching matrices and is also **NOT expected to generalize** to unseen graph pairs. It only aims to provide a relatively good training signal that guides the matching model within the current training epoch. Similar to those discriminators used in image generation, they are not intended to serve as a useful or generalizable image classifier after training.
>
> Therefore, we only need to evaluate **discriminator's online performance**, and an reasonable evaluation is to see **whether it can help the generator to explore better solutions more efficiently during training** (this is also what we only really need to care about). This is demonstrated in Figure 3 of our paper, the discriminator is very effective in helping the generator to find near-optimal solutions for training data.
>
> An alternative evaluation is to show **the loss curve of the discriminator**, which is an indicator of the correlation between the discriminator's preference and true edit distance during training. However, we are **not allowed to post the any visualization results** during the rebuttal period, we will show it in the camera-ready version.
>
>
> ---
>
> ### **2. Evaluation of the training cost**
> Again, please note that, the discriminator is **trained in an online manner** with the matching model using the latest sampled data, all performance results of our GEDRanker are trained with **the same number of epochs** as supervised learning methods. And the results demonstrate that our GEDRanker can achieve **similar performance as supervised learning method with the same number of epochs**. This is mainly due to the design of the preference-aware GAN, which is our core contribution.
>
> But we do have extra training cost per epoch, the extra training cost comes from **the sampling of solutions** and **the forward/backward pass of the discriminator**. But note that, the extra training cost of the discriminator **only occurs in the exploration phase**, and the discriminator is lighter than the generator (less number of parameters). Below is the training cost per epoch of exploration stage and exploitation stage for our unsupervised approach:
>
> | Dataset | Exploration Stage | Exploitation Stage |
> |---------|-------------------|--------------------|
> | AIDS    | 58s | 38s |
> | Linux   | 1m55s | 1m12s |
> | IMDB    | 3m11s | 2m07s |
>
> Below is the training cost per epoch for supervised learning:
> | Dataset |  |
> |---------|--------------|
> | AIDS    | 15s |
> | Linux   | 28s |
> | IMDB    | 53s |
>
> During the exploration stage, the additional time cost arises from both the discriminator and the solution sampling. While the cost of solution sampling is unavoidable in unsupervised learning, it could be further reduced by efficient C++ implementation. For the discriminator, in our implementation we did not parallelize multiple forward passes (line 6 of Algorithm 3) due to hardware limitations. With a more powerful machine, this cost **could be further reduced through parallelization**, ideally cutting the discriminator-related time by half.
>
> During the exploitation stage, the addtional time cost arises from the solution sampling only.
>
> And note that the cost of the generator in unsupervised learning is the same as in supervised learning.
>
> ---
>
> ### **3. Evaluation of semi-supervised learning**
> Regarding the performance of semi-supervised case, we perform experiments by initializing 30% of total training graph pairs' best solutions using the ground-truth solutions. Below are the results of GEDRanker under semi-supervised setting:
>
> |Method| Dataset | MAE   | Accuracy    | ρ     | τ     | pk10  | pk20  |
> |----|---------|-------|--------|-------|-------|-------|-------|
> |Semi-supervised| AIDS    | 0.062 | 94.6%  | 0.989 | 0.978 | 99.2% | 99.2% |
> || Linux   | 0.003 | 99.8%  | 0.999 | 0.998 | 100%  | 99.8% |
> || IMDB    | 1.229 | 92.8%  | 0.965 | 0.955 | 96.3% | 96.9% |
> |Unsupervised|AIDS|0.088|92.6%|0.984|0.969|99.1%|99.1%|
> ||Linux|0.01|99.5%|0.997|0.995|100%|99.8%|
> ||IMDB|1.019|94%|0.999|0.97|96.1%|97%|
>
> For the AIDS and Linux datasets, semi-supervised learning with 30% ground-truth slightly improves performance compared to unsupervised learning without any ground-truth. For IMDB dataset, the results are evaluated based on the **approximated** synethic ground-truth (might be larger than the exact ground-truth), therefore they only indicate the distance between the predicted GED and the **approximated** ground-truth GED (the predicted GED could be better than the approximated ground-truth GED).
>
> ---
>
> ### **4. Matching initalization**
> A good initialization of the current best solution can indeed reduce the number of epochs required for exploration. However, obtaining initial solutions for training data using ** GEDGW is computationally expensive**. As shown in Table 1, **GEDGW requires at least $2-6\times$ longer inference time** compared to GEDRanker and DiffGED, where the latter two are evaluated with **10 forward passes** of the network. The gap becomes even larger during training, as GEDRanker and DiffGED **only require a single forward pass** of the matching network, making **GEDGW at least $20-60\times$ slower**. And the number of training data is more than the number of testing data. Therefore, **using GEDGW as an initialization for training data is too costly in practice**. Furthermore, since GEDGW is NOT an exact algorithm and only provides **sub-optimal solutions**, initializing the best solution with GEDGW could **bias GEDRanker to explore within a sub-optimal region**. In our framework, we simply use randomly initialized solutions, and Figure 3 shows that even with random initialization, GEDRanker can still efficiently converge to near-optimal solutions.
>
>
> ---
>
> ### **5.Scalability**
>
> We would like to clarify that, for IMDB-large dataset, we **don't know what the optimal GED is**, so we cannot know whether if both of GEDRanker and GEDGW are just close to the optimal results, hence resulting in close performance. But our method can find **a smaller GED and with a shorter time**. It is worth noting that GEDGW is an **optimization** algorithm, meaning its performance cannot be further improved easily **without extra time cost on the test instances** (and some of them can only output a fixed approximation). In contrast, neural methods **offer the potential for performance improvement** through increased training data, longer training epochs, or other techniques, these do not require extra time cost on the test instances. Therefore, comparison with the optimization algorithms is **NOT the primary focus** of this work.
>
> More importantly, please note that **the scalability can be inherently influenced by the base model**, DiffGED. Therefore, we think a more appropriate way to demonstrate our method's effectiveness on large graphs is by evaluating **how it enhances the scalability of the base model in practice**. To this end, we train the base model (DiffGED) only on small real-world IMDB graph pairs where ground-truth labels are available. We then evaluate it on the IMDB-large graph pairs used in Table 5 to see the scalability of the original base model, and compared with our GEDRanker that enables the training of base model on large graph pairs without requiring ground-truth supervision. Following are the evaluation results:
>
> | Method     | Average GED | Time(s) |
> |------------|-------------|---------|
> | DiffGED    | 143.607      | 0.1911  |
> | GEDGW      | 140.278      | 0.4032  |
> | GEDRanker  | 140.138      | 0.1904  |
>
> Under this practical setting, we can see that the supervised DiffGED underperforms the optimization algorithm GEDGW in terms of average GED. However, by leveraging our unsupervised training strategy (GEDRanker), DiffGED can be trained on large graphs and subsequently outperforms GEDGW in both average GED and running time.

---

> > ### Comment · Reviewer_eHfB · 2025-08-05
> > **Thank you**
> >
> > Thank you for the detailed explanation and additional experimental evaluation.

---

### Decision · Program_Chairs · 2025-09-17

**Decision:**

Accept (poster)

**Comment:**

This paper presents an unsupervised framework for graph node matching and Graph Edit Distance (GED) estimation, designed to avoid reliance on ground-truth GED labels, which are expensive to collect. The model uses a diffusion-based backbone to generate node mapping matrices and derives edit paths from them. Instead of supervised matching, training proceeds by engaging a preference-aware discriminator. The discriminator evaluates mappings, ensuring that shorter edit paths are favored over longer ones. This GAN-inspired setup provides a more interpretable objective and balances exploration with exploitation during training. Experimental results show that GEDRanker achieves near-optimal GED solutions without supervision, outperforming traditional approximate methods and approaching the accuracy and efficiency of supervised state-of-the-art models.

This is a useful and competent submission that is certainly a poster accept.  The novelty of unsupervised GED is somewhat overstated. Y63G has suggested a better bibliography against which to position the work.  Apart from those works, the paper itself states that supervised diffusion-based node matching models [11] already exist, although the removal of supervision from this family of approaches is a worthy contribution.

Otherwise, reviewers had fairly routine questions about notation, definitions, and experimental details, which were largely adequately addressed during rebuttal.

A couple of additional writing issues:

L153 Given there is no need for supervision, it is not immediately clear how the training graph pairs are collected or sampled.  Presumably a training graph pair is offered as-is, without any GED value or correspondence as supervision.

L166 caption "rank the preference over" is confusing language.  Reword.

Since the title says "preference-aware"  and the abstract says "without the need for ground-truth labels", the abstract itself should clearly state that no supervision is needed, and what data is needed from outside during training, vs. what can be generated internally.